# Nearly Tight Bounds for Robust Proper Learning of Halfspaces with a Margin[*]

**Ilias Diakonikolas**
University of Wisconsin-Madison
ilias@cs.wisc.edu

**Daniel M. Kane**
University of California, San Diego
dakane@cs.ucsd.edu

**Pasin Manurangsi**[†]
University of California, Berkeley
pasin@berkeley.edu

## Abstract

We study the problem of *properly* learning large margin halfspaces in the agnostic PAC model. In more detail, we study the complexity of properly learning $d$-dimensional halfspaces on the unit ball within misclassification error $\alpha \cdot \mathrm{OPT}_\gamma + \epsilon$, where $\mathrm{OPT}_\gamma$ is the optimal $\gamma$-margin error rate and $\alpha \geq 1$ is the approximation ratio. We give learning algorithms and computational hardness results for this problem, for all values of the approximation ratio $\alpha \geq 1$, that are nearly-matching for a range of parameters. Specifically, for the natural setting that $\alpha$ is any constant bigger than one, we provide an essentially tight complexity characterization. On the positive side, we give an $\alpha = 1.01$-approximate proper learner that uses $O(1/(\epsilon^2\gamma^2))$ samples (which is optimal) and runs in time $\mathrm{poly}(d/\epsilon) \cdot 2^{\tilde{O}(1/\gamma^2)}$. On the negative side, we show that *any* constant factor approximate proper learner has runtime $\mathrm{poly}(d/\epsilon) \cdot 2^{(1/\gamma)^{2-o(1)}}$, assuming the Exponential Time Hypothesis.

## 1 Introduction

### 1.1 Background and Problem Definition

Halfspaces are boolean functions $h_{\mathbf{w}} : \mathbb{R}^d \to \{\pm 1\}$ of the form $h_{\mathbf{w}}(\mathbf{x}) = \mathrm{sign}\left(\langle \mathbf{w}, \mathbf{x} \rangle\right)$, where $\mathbf{w} \in \mathbb{R}^d$ is the associated weight vector. (The function $\mathrm{sign} : \mathbb{R} \to \{\pm 1\}$ is defined as $\mathrm{sign}(u) = 1$ if $u \geq 0$ and $\mathrm{sign}(u) = -1$ otherwise.) The problem of learning an unknown halfspace with a margin condition (in the sense that no example is allowed to lie too close to the separating hyperplane) is as old as the field of machine learning — starting with Rosenblatt's Perceptron algorithm [Ros58] — and has arguably been one of the most influential problems in the development of the field, with techniques such as SVMs [Vap98] and AdaBoost [FS97] coming out of its study.

We study the problem of learning $\gamma$-margin halfspaces in the *agnostic* PAC model [Hau92, KSS94]. Specifically, there is an unknown distribution $\mathcal{D}$ on $\mathbb{B}_d \times \{\pm 1\}$, where $\mathbb{B}_d$ is the unit ball on $\mathbb{R}^d$, and the learning algorithm $\mathcal{A}$ is given as input a training set $S = \{(\mathbf{x}^{(i)}, y^{(i)})\}_{i=1}^m$ of i.i.d. samples drawn from $\mathcal{D}$. The goal of $\mathcal{A}$ is to output a hypothesis whose error rate is competitive with the $\gamma$-margin error rate of the optimal halfspace. In more detail, the *error rate* (misclassification error) of a hypothesis $h : \mathbb{R}^d \to \{\pm 1\}$ (with respect to $\mathcal{D}$) is $\mathrm{err}_{0-1}^{\mathcal{D}}(h) \stackrel{\text{def}}{=} \mathbf{Pr}_{(\mathbf{x},y)\sim\mathcal{D}}[h(\mathbf{x}) \neq y]$. For $\gamma \in (0,1)$, the $\gamma$-*margin error rate* of a halfspace $h_{\mathbf{w}}(\mathbf{x})$ with $\|\mathbf{w}\|_2 \leq 1$ is $\mathrm{err}_\gamma^{\mathcal{D}}(\mathbf{w}) \stackrel{\text{def}}{=} \mathbf{Pr}_{(\mathbf{x},y)\sim\mathcal{D}}[y\langle \mathbf{w}, x \rangle \leq \gamma]$.

---

[*]The full version of this paper is available at [DKM19].

[†]Now at Google Research, Mountain View.

We denote by $\mathrm{OPT}_\gamma^{\mathcal{D}} \stackrel{\text{def}}{=} \min_{\|\mathbf{w}\|_2 \leq 1} \mathrm{err}_\gamma^{\mathcal{D}}(\mathbf{w})$ the minimum $\gamma$-margin error rate achievable by any halfspace. We say that $\mathcal{A}$ is an $\alpha$-*agnostic learner*, $\alpha \geq 1$, if it outputs a hypothesis $h$ that with probability at least $1 - \tau$ satisfies $\mathrm{err}_{0-1}^{\mathcal{D}}(h) \leq \alpha \cdot \mathrm{OPT}_\gamma^{\mathcal{D}} + \epsilon$. (For $\alpha = 1$, we obtain the standard notion of agnostic learning.) If the hypothesis $h$ is itself a halfspace, we say that the learning algorithm is *proper*. This work focuses on proper learning algorithms.

## 1.2 Related and Prior Work

In this section, we summarize the related work that is directly relevant to the results of this paper. First, we note that the sample complexity of our learning problem (ignoring computational considerations) is well-understood. In particular, the ERM that minimizes the number of $\gamma$-*margin errors* over the training set (subject to a norm constraint) is known to be an agnostic learner ($\alpha = 1$), assuming the sample size is $\Omega(\log(1/\tau)/(\epsilon^2\gamma^2))$. Specifically, $\Theta(\log(1/\tau)/(\epsilon^2\gamma^2))$ samples[3] are known to be sufficient and necessary for this learning problem (see, e.g., [BM02, McA03]). In the realizable case ($\mathrm{OPT}_\gamma^{\mathcal{D}} = 0$), i.e., if the data is linearly separable with margin $\gamma$, the ERM rule above can be implemented in $\mathrm{poly}(d, 1/\epsilon, 1/\gamma)$ time using the Perceptron algorithm. The agnostic setting ($\mathrm{OPT}_\gamma^{\mathcal{D}} > 0$) is much more challenging computationally.

The agnostic version of our problem ($\alpha = 1$) was first considered in [BS00], who gave a *proper* learning algorithm with runtime $\mathrm{poly}(d) \cdot (1/\epsilon)^{\tilde{O}(1/\gamma^2)}$. It was also shown in [BS00] that agnostic proper learning with runtime $\mathrm{poly}(d, 1/\epsilon, 1/\gamma)$ is NP-hard. A question left open by their work was characterizing the computational complexity of proper learning as a function of $1/\gamma$.

Subsequent works focused on *improper* learning. The $\alpha = 1$ case was studied in [SSS09, SSS10] who gave a learning algorithm with sample complexity $\mathrm{poly}(1/\epsilon) \cdot 2^{\tilde{O}(1/\gamma)}$ – i.e., *exponential* in $1/\gamma$ – and computational complexity $\mathrm{poly}(d/\epsilon) \cdot 2^{\tilde{O}(1/\gamma)}$. The increased sample complexity is inherent in their approach, as their algorithm works by solving a convex program over an expanded feature space. [BS12] gave an $\alpha$-agnostic learning algorithm for all $\alpha \geq 1$ with sample complexity $2^{\tilde{O}(1/(\alpha\gamma))}$ and computational complexity $\mathrm{poly}(d/\epsilon) \cdot 2^{\tilde{O}(1/(\alpha\gamma))}$. (We note that the Perceptron algorithm is known to achieve $\alpha = 1/\gamma$ [Ser01]. Prior to [BS12], [LS11] gave a $\mathrm{poly}(d, 1/\epsilon, 1/\gamma)$ time algorithm achieving $\alpha = \Theta((1/\gamma)/\sqrt{\log(1/\gamma)})$.) [BS12] posed as an open question whether their upper bounds for improper learning can be also derived for a proper learner.

A related line of work [KLS09, ABL17, DKK+16, LRV16, DKK+17, DKK+18, DKS18, KKM18, DKS19, DKK+19] has given polynomial time robust estimators for a range of learning tasks. Specifically, [KLS09, ABL17, DKS18, DKK+19] obtained efficient PAC learning algorithms for halfspaces with malicious noise [Val85, KL93], under the assumption that the uncorrupted data comes from a "tame" distribution, e.g., Gaussian or isotropic log-concave. It should be noted that the class of $\gamma$-margin distributions considered in this work is significantly broader and can be far from satisfying the structural properties required in the aforementioned works.

A growing body of theoretical work has focused on *adversarially robust learning* (e.g., [BLPR19, MHS19, DNV19, Nak19]). In adversarially robust learning, the learner seeks to output a hypothesis with small $\gamma$-*robust misclassification error*, which for a hypothesis $h$ and a norm $\|\cdot\|$ is typically defined as $\mathbf{Pr}_{(\mathbf{x},y)\sim\mathcal{D}}[\exists \mathbf{x}' \text{ with } \|\mathbf{x}' - \mathbf{x}\| \leq \gamma \text{ s.t. } h(\mathbf{x}') \neq y]$. Notice that when $h$ is a halfspace and $\|\cdot\|$ is the Euclidean norm, the $\gamma$-robust misclassification error coincides with the $\gamma$-margin error in our context. (It should be noted that most of the literature on adversarially robust learning focuses on the $\ell_\infty$-norm.) However, the objectives of the two learning settings are slightly different: in adversarially robust learning, the learner would like to output a hypothesis with small $\gamma$-robust misclassification error, whereas in our context the learner only has to output a hypothesis with small zero-one misclassification error. Nonetheless, as we point out in Remark 1.3, our algorithms can be adapted to provide guarantees in line with the adversarially robust setting as well.

Finally, in the distribution-independent agnostic setting without margin assumptions, there is compelling complexity-theoretic evidence that even weak learning of halfspaces is computationally intractable [GR06, FGKP06, DOSW11, Dan16, BGS18].

## 1.3 Our Contributions

We study the complexity of *proper* $\alpha$-agnostic learning of $\gamma$-margin halfspaces on the unit ball. Our main result nearly characterizes the complexity of constant factor approximation to this problem:

**Theorem 1.1.** *There is an algorithm that uses $O(1/(\epsilon^2\gamma^2))$ samples, runs in time $\mathrm{poly}(d/\epsilon)\cdot 2^{\tilde{O}(1/\gamma^2)}$ and is an $\alpha = 1.01$-agnostic proper learner for $\gamma$-margin halfspaces with confidence probability $9/10$. Moreover, assuming the randomized Exponential Time Hypothesis, any proper learning algorithm that achieves any constant factor approximation has runtime $\mathrm{poly}(d/\epsilon)\cdot\Omega(2^{1/\gamma^{2-o(1)}})$.*

The reader is referred to Theorem 2.1 for the upper bound and Theorem 3.1 for the lower bound. First, we note that the approximation ratio of $1.01$ in the theorem statement is not inherent. Our algorithm achieves $\alpha = 1 + \delta$, for any $\delta > 0$, with runtime $\mathrm{poly}(d/\epsilon)\cdot 2^{\tilde{O}(1/(\delta\gamma^2))}$. The runtime of our algorithm significantly improves the runtime of the best known agnostic proper learner [BS00], achieving fixed polynomial dependence on $1/\epsilon$, independent of $\gamma$. This gain in runtime comes at the expense of losing a small constant factor in the error guarantee. It is natural to ask whether there exists an 1-agnostic proper learner matching the runtime of our Theorem 1.1. In Theorem 3.2, we establish a computational hardness result implying that such an improvement is unlikely.

The runtime dependence of our algorithm scales as $2^{\tilde{O}(1/\gamma^2)}$ (which is nearly best possible for proper learners), as opposed to $2^{\tilde{O}(1/\gamma)}$ for improper learning [SSS09, BS12]. In addition to the interpretability of proper learning, the sample complexity of our algorithm is quadratic in $1/\gamma$ (which is optimal), as opposed to exponential for known improper learners. Moreover, for moderate values of $\gamma$, our algorithm may be faster than known improper learners, as it only uses spectral methods and ERM, as opposed to convex programming. Finally, we note that the lower bound part of Theorem 1.1 implies a computational separation between proper and improper learning for this problem.

In addition, we explore the complexity of $\alpha$-agnostic learning for large $\alpha > 1$. The following theorem summarizes our results in this setting:

**Theorem 1.2.** *There is an algorithm that uses $\tilde{O}(1/(\epsilon^2\gamma^2))$ samples, runs in time $\mathrm{poly}(d)\cdot(1/\epsilon)^{\tilde{O}(1/(\alpha\gamma)^2)}$ and is a $\alpha$-agnostic proper learner for $\gamma$-margin halfspaces with confidence probability $9/10$. Moreover, assuming $\mathrm{NP}\neq\mathrm{RP}$ and the Sliding Scale Conjecture [BGLR94], no $(1/\gamma)^c$-agnostic proper learner runs in $\mathrm{poly}(d,1/\varepsilon,1/\gamma)$ time for some (absolute) constant $c > 0$.*

The reader is referred to Theorem 3.3 for a more precise statement of the lower bound; the upper bound is deferred to the full version [DKM19]. In summary, we give an $\alpha$-agnostic algorithm with runtime exponential in $1/(\alpha\gamma)^2$, as opposed to $1/\gamma^2$, and we show that achieving $\alpha = (1/\gamma)^{\Omega(1)}$ is computationally hard. (Assuming only $\mathrm{NP}\neq\mathrm{RP}$, we can rule out polynomial time $\alpha$-agnostic proper learning for $\alpha = (1/\gamma)^{\frac{1}{\mathrm{polyloglog}(1/\gamma)}}$.)

**Remark 1.3.** While not stated explicitly in the subsequent analysis, our algorithms (with a slight modification to the associated constant factors) not only give a halfspace $\mathbf{w}^*$ with zero-one loss at most $\alpha\cdot\mathrm{OPT}_\gamma^{\mathcal{D}} + \epsilon$, but this guarantee holds for the $0.99\gamma$-margin error[4] of $\mathbf{w}^*$ as well. Thus, our learning algorithms also work in the adversarially robust setting (under the Euclidean norm) with a small loss in the "robustness parameter" (margin) from the one used to compute the optimum (i.e., $\gamma$) to the one used to measure the error of the output hypothesis (i.e., $0.99\gamma$).

## 1.4 Our Techniques

**Overview of Algorithms.** For the sake of this intuitive explanation, we provide an overview of our algorithms when the underlying distribution $\mathcal{D}$ is explicitly known. The finite sample analysis of our algorithms follows from standard generalization bounds (see Section 2).

Our constant factor approximation algorithm relies on the following observation: Let $\mathbf{w}^*$ be the optimal weight vector. The assumption that $|\langle\mathbf{w}^*,\mathbf{x}\rangle|$ is large for almost all $\mathbf{x}$ (by the margin property), implies a relatively strong condition on $\mathbf{w}^*$, which will allow us to find a relatively small search space for a near-optimal solution. A first idea is to consider the matrix $\mathbf{M} = \mathbf{E}_{(\mathbf{x},y)\sim\mathcal{D}}[\mathbf{x}\mathbf{x}^T]$, and note that

$\mathbf{w}^{*T}\mathbf{M}\mathbf{w}^* = \Omega(\gamma^2)$. This in turn implies that $\mathbf{w}^*$ has a large component on the subspace spanned by the largest $O(1/(\epsilon\gamma^2))$ eigenvalues of $\mathbf{M}$. This idea suggests a basic algorithm that computes a net over unit-norm weight vectors on this subspace and outputs the best answer. Unfortunately, this basic algorithm has runtime $\text{poly}(d)2^{\tilde{O}(1/(\epsilon\gamma^2))}$. (Details are deferred to the full version [DKM19].)

To obtain our $\text{poly}(d/\epsilon)2^{\tilde{O}(1/\gamma^2)}$ time constant factor approximation algorithm (Theorem 1.1), we use a refinement of the above idea. Instead of trying to guess the projection of $\mathbf{w}^*$ onto the space of large eigenvectors *all at once*, we will do so in stages. In particular, it is not hard to see that $\mathbf{w}^*$ has a non-trivial projection onto the subspace spanned by the top $O(1/\gamma^2)$ eigenvalues of $\mathbf{M}$. If we guess this projection, we will have some approximation to $\mathbf{w}^*$, but unfortunately not a sufficiently good one. However, we note that the difference between $\mathbf{w}^*$ and our current hypothesis $\mathbf{w}$ will have a large average squared inner product with the misclassified points. This suggests an iterative algorithm that in the $i$-th iteration considers the second moment matrix $\mathbf{M}^{(i)}$ of the points not correctly classified by the current hypothesis $\text{sign}(\langle\mathbf{w}^{(i)},\mathbf{x}\rangle)$, guesses a vector $\mathbf{u}$ in the space spanned by the top few eigenvalues of $\mathbf{M}^{(i)}$, and sets $\mathbf{w}^{(i+1)} = \mathbf{u} + \mathbf{w}^{(i)}$. This procedure produces a candidate set of weights with cardinality $2^{\tilde{O}(1/\gamma^2)}$ one of which has the desired misclassification error. This algorithm and its analysis are given in Section 2.

Our general $\alpha$-factor algorithm (Theorem 1.2) relies on approximating the *Chow parameters* of the target halfspace $f$, i.e., the $d$ numbers $\mathbf{E}[f(\mathbf{x})\mathbf{x}_i]$, $i \in [d]$. A classical result [Cho61] shows that the exact values of the Chow parameters of a halfspace (over any distribution) uniquely define the halfspace. Although this uniqueness is not very useful in general, the margin assumption implies a relatively strong approximate identifiability result. Combining this with an algorithm of [DDFS14], we can efficiently compute an approximation to the halfspace $f$ given an approximation to its Chow parameters. In particular, if we can approximate the Chow parameters to $\ell_2$-error $\nu \cdot \gamma$, we can approximate $f_{\mathbf{w}^*}$ within error $\text{OPT}_\gamma^{\mathcal{D}} + \nu$.

The natural way to approximate the Chow parameters would be by computing the empirical Chow parameters, namely $\mathbf{E}_{(\mathbf{x},y)\sim\mathcal{D}}[y\mathbf{x}]$. In the realizable case, this quantity corresponds exactly to the vector of Chow parameters. Unfortunately, this does not work in the agnostic case and it can introduce an error of $\omega(\text{OPT}_\gamma^{\mathcal{D}})$. To overcome this obstacle, we note that in order for a small fraction of errors to introduce a large error in the empirical Chow parameters, it must be the case that there is some direction $\mathbf{w}$ in which many of these erroneous points introduce a large error. If we can guess some error that correlates well with $\mathbf{w}$ and also guess the correct projection of our Chow parameters onto this vector, we can correct a decent fraction of the error between the empirical and true Chow parameters. We show that making the correct guesses $\tilde{O}(1/(\gamma\alpha)^2)$ times, we can reduce the empirical error sufficiently so that it can be used to find an accurate hypothesis. Once again, we can compute a hypothesis for each sequence of guesses and return the best one. The formal description of the algorithm and its analysis can be found in the full version of this paper [DKM19].

**Overview of Computational Lower Bounds.** Our hardness results are shown via two reductions. These reductions take in an instance of a "hard problem" and produce a distribution $\mathcal{D}$ on $\mathbb{B}_d \times \{\pm 1\}$. If the starting instance is a YES instance of the original problem, then $\text{OPT}_\gamma^{\mathcal{D}}$ is small for an appropriate $\gamma$. On the other hand, if the starting instance is a NO instance of the original problem, then $\text{OPT}_{0-1}^{\mathcal{D}}$ is large[5]. As a result, if there is a "too fast" ($\alpha$-)agnostic learner for $\gamma$-margin halfspaces, then we would also get a "too fast" algorithm for the starting problem as well, which would violate the corresponding complexity assumption.

To understand the margin parameter $\gamma$ we can achieve, we need to first understand the problems we start with. For our reductions, the starting problems can be viewed in the following form: select $k$ items from $v_1, \ldots, v_N$ that satisfy certain "local constraints". For instance, in our first reduction, the reduction is from the $k$-Clique problem where we are given a graph $G$ and an integer $k$, and the goal is to determine whether it contains a $k$-clique as a subgraph. For this problem, $v_1, \ldots, v_N$ are the vertices of $G$ and the "local" constraints are that every pair of selected vertices induces an edge.

Roughly speaking, our reduction produces $\mathcal{D}$ with dimension $d = N$, with the $i$-th dimension corresponding to $v_i$. The "ideal" solution in the YES case is to set $\mathbf{w}_i = \frac{1}{\sqrt{k}}$ iff $v_i$ is selected and set $\mathbf{w}_i = 0$ otherwise. In our reductions, the local constraints are expressed using "sparse" sample vectors

(with only a constant number of non-zero coordinates all having the same magnitude). For example, in the case of $k$-Clique, the constraints can be expressed as: for every non-edge $(i, j)$, we must have $\left(\frac{1}{\sqrt{2}}\mathbf{e}^i + \frac{1}{\sqrt{2}}\mathbf{e}^j\right) \cdot \mathbf{w} \leq \frac{1}{\sqrt{2k}}$, where $\mathbf{e}^i$ and $\mathbf{e}^j$ denote the $i$-th and $j$-th vectors in the standard basis. A main step in both of our proofs is to show that the reduction still works even when we "shift" the right hand side by a small multiple of $\frac{1}{\sqrt{k}}$. For instance, in the case of $k$-Clique, it is possible to show that, even if we replace $\frac{1}{\sqrt{2k}}$ with say $\frac{0.99}{\sqrt{2k}}$, the correctness of the construction remains and we also get the added benefit that now the constraints are satisfied by a margin of $\gamma = \Theta(\frac{1}{\sqrt{k}})$ for our ideal solution in the YES case.

In the case of $k$-Clique, the above idea yields a reduction to 1-agnostic learning $\gamma$-margin halfspaces with margin $\gamma = \Theta(\frac{1}{\sqrt{k}})$, where the dimension $d$ is $N$ (and $\varepsilon = \frac{1}{\text{poly}(N)}$). As a result, if there is an $f(\frac{1}{\gamma})\text{poly}(d, \frac{1}{\varepsilon})$-time algorithm for the latter for some function $f$, then there also exists an $g(k)\text{poly}(N)$-time algorithm for $k$-Clique for some function $g$, which is considered unlikely as it would break a widely-believed hypothesis in the area of parameterized complexity.

Ruling out $\alpha$-agnostic learners is slightly more complicated, since we need to produce the "gap" of $\alpha$ between $\text{OPT}_\gamma^{\mathcal{D}}$ in the YES case and $\text{OPT}_{0-1}^{\mathcal{D}}$ in the NO case. To create such a gap, we appeal to the PCP Theorem [AS98, ALM+98] which can be thought of as an NP-hardness proof of the following "gap version" of 3SAT: given a 3CNF formula as input, distinguish between the case where the formula is satisfiable and the case where the formula is not even 0.9-satisfiable[6]. Moreover, further strengthened versions of the PCP Theorem [Din07, MR10] actually implies that this Gap-3SAT problem cannot even be solved in time $O(2^{n^{0.999}})$, where $n$ denotes the number of variables in the formula, assuming the exponential time hypothesis (ETH)[7]. Once again, (Gap-)3SAT can be viewed in the form of "item selection with local constraint". However, the number of selected items $k$ is now equal to $n$, the number of variables of the formula. With a similar line of reasoning as above, the margin we get is now $\gamma = \Theta(\frac{1}{\sqrt{k}}) = \Theta(\frac{1}{\sqrt{n}})$. As a result, if there is a say $2^{(1/\gamma)^{1.99}}\text{poly}(d, \frac{1}{\varepsilon})$-time $\alpha$-agnostic learner for $\gamma$-margin halfspaces (for an appropriate $\alpha$), then there is an $O(2^{n^{0.995}})$-time algorithm for Gap-3SAT, which would violate ETH.

Unfortunately, the above described idea only gives the "gap" $\alpha$ that is only slightly larger than 1, because the gap that we start with in the Gap-3SAT problem is already pretty small. To achieve larger gaps, our actual reduction starts from a generalization of 3SAT called constraint satisfaction problems (CSPs), whose gap problems are hard even for very large gap. This concludes the outline of the main intuitions in our reductions.

## 1.5 Preliminaries

For $n \in \mathbb{Z}_+$, we denote $[n] \overset{\text{def}}{=} \{1, \ldots, n\}$. We will use small boldface characters for vectors and capital boldface characters for matrices. For a vector $\mathbf{x} \in \mathbb{R}^d$, and $i \in [d]$, $\mathbf{x}_i$ denotes the $i$-th coordinate of $\mathbf{x}$, and $\|\mathbf{x}\|_2 \overset{\text{def}}{=} (\sum_{i=1}^d \mathbf{x}_i^2)^{1/2}$ denotes the $\ell_2$-norm of $\mathbf{x}$. We will use $\langle \mathbf{x}, \mathbf{y} \rangle$ for the inner product between $\mathbf{x}, \mathbf{y} \in \mathbb{R}^d$. For a matrix $\mathbf{M} \in \mathbb{R}^{d \times d}$, we will denote by $\|\mathbf{M}\|_2$ its spectral norm and by $\text{tr}(\mathbf{M})$ its trace. Let $\mathbb{B}_d = \{\mathbf{x} \in \mathbb{R}^d : \|\mathbf{x}\|_2 \leq 1\}$ be the unit ball and $\mathbb{S}_{d-1} = \{\mathbf{x} \in \mathbb{R}^d : \|\mathbf{x}\|_2 = 1\}$ be the unit sphere in $\mathbb{R}^d$. An origin-centered halfspace is a Boolean-valued function $h_\mathbf{w} : \mathbb{R}^d \to \{\pm 1\}$ of the form $h_\mathbf{w}(\mathbf{x}) = \text{sign}(\langle \mathbf{w}, \mathbf{x} \rangle)$, where $\mathbf{w} \in \mathbb{R}^d$. (Note that we may assume w.l.o.g. that $\|\mathbf{w}\|_2 = 1$.) Let $\mathcal{H}_d = \{h_\mathbf{w}(\mathbf{x}) = \text{sign}(\langle \mathbf{w}, \mathbf{x} \rangle), \mathbf{w} \in \mathbb{R}^d\}$ denote the class of all origin-centered halfspaces on $\mathbb{R}^d$. Finally, we use $\mathbf{e}^i$ to denote the $i$-th standard basis vector, i.e., the vector whose $i$-th coordinate is one and the remaining coordinates are zeros.

## 2 Algorithm for Proper Agnostic Learning of Halfspaces with a Margin

In this section, we show the following theorem, which gives the upper bound part of Theorem 1.1:

**Theorem 2.1.** *Fix $0 < \delta \leq 1$. There is an algorithm that uses $O(1/(\epsilon^2 \gamma^2))$ samples, runs in time $\mathrm{poly}(d/\epsilon) \cdot 2^{\tilde{O}(1/(\delta \gamma^2))}$ and is a $(1 + \delta)$-agnostic proper learner for $\gamma$-margin halfspaces with confidence probability $9/10$.*

Our algorithm in this section produces a finite set of candidate weight vectors and outputs the one with the smallest empirical $\gamma/2$-margin error. For the sake of this intuitive description, we will assume that the algorithm knows the distribution $\mathcal{D}$ in question supported on $\mathbb{B}_d \times \{\pm 1\}$. By assumption, there is a unit vector $\mathbf{w}^*$ so that $\mathrm{err}_\gamma^{\mathcal{D}}(\mathbf{w}^*) \leq \mathrm{OPT}_\gamma^{\mathcal{D}}$.

We note that if a hypothesis $h_{\mathbf{w}}$ defined by vector $\mathbf{w}$ has $\gamma/2$-margin error at least a $(1+\delta)\mathrm{OPT}_\gamma^{\mathcal{D}}$, then there must be a large number of points correctly classified with $\gamma$-margin by $h_{\mathbf{w}^*}$, but not correctly classified with $\gamma/2$-margin by $h_{\mathbf{w}}$. For all of these points, we must have that $|\langle \mathbf{w}^* - \mathbf{w}, \mathbf{x} \rangle| \geq \gamma/2$. This implies that the $\gamma/2$-margin-misclassified points of $h_{\mathbf{w}}$ have a large covariance in the $\mathbf{w}^* - \mathbf{w}$ direction. In particular, we have:

**Claim 2.2.** *Let $\mathbf{w} \in \mathbb{R}^d$ be such that $\mathrm{err}_{\gamma/2}^{\mathcal{D}}(\mathbf{w}) > (1 + \delta)\mathrm{OPT}_\gamma^{\mathcal{D}}$. Let $\mathcal{D}'$ be $\mathcal{D}$ conditioned on $y\langle \mathbf{w}, \mathbf{x} \rangle \leq \gamma/2$. Let $\mathbf{M}^{\mathcal{D}'} = \mathbf{E}_{(\mathbf{x},y) \sim \mathcal{D}'}[\mathbf{x}\mathbf{x}^T]$. Then $(\mathbf{w}^* - \mathbf{w})^T \mathbf{M}^{\mathcal{D}'}(\mathbf{w}^* - \mathbf{w}) \geq \delta \gamma^2/8$.*

*Proof.* We claim that with probability at least $\delta/2$ over $(\mathbf{x}, y) \sim \mathcal{D}'$ we have that $y\langle \mathbf{w}, \mathbf{x} \rangle \leq \gamma/2$ and $y\langle \mathbf{w}^*, \mathbf{x} \rangle \geq \gamma$. To see this, we first note that $\mathbf{Pr}_{(\mathbf{x},y) \sim \mathcal{D}'}[y\langle \mathbf{w}, \mathbf{x} \rangle > \gamma/2] = 0$ holds by definition of $\mathcal{D}'$. Hence, we have that

$$\mathbf{Pr}_{(\mathbf{x},y) \sim \mathcal{D}'}[y\langle \mathbf{w}^*, \mathbf{x} \rangle \leq \gamma] \leq \frac{\mathbf{Pr}_{(\mathbf{x},y) \sim \mathcal{D}}[y\langle \mathbf{w}^*, \mathbf{x} \rangle \leq \gamma]}{\mathbf{Pr}_{(\mathbf{x},y) \sim \mathcal{D}}[y\langle \mathbf{w}, \mathbf{x} \rangle \leq \gamma/2]} < \frac{\mathrm{OPT}_\gamma^{\mathcal{D}}}{(1 + \delta)\mathrm{OPT}_\gamma^{\mathcal{D}}} = \frac{1}{(1 + \delta)} \ .$$

By a union bound, we obtain $\mathbf{Pr}_{(\mathbf{x},y) \sim \mathcal{D}'}[(y\langle \mathbf{w}, \mathbf{x} \rangle > \gamma/2) \cup (y\langle \mathbf{w}^*, \mathbf{x} \rangle \leq \gamma)] \leq \frac{1}{(1+\delta)}$.

Therefore, with probability at least $\delta/(1 + \delta) \geq \delta/2$ (since $\delta \leq 1$) over $(\mathbf{x}, y) \sim \mathcal{D}'$ we have that $y\langle \mathbf{w}^* - \mathbf{w}, \mathbf{x} \rangle \geq \gamma/2$, which implies that $\langle \mathbf{w}^* - \mathbf{w}, \mathbf{x} \rangle^2 \geq \gamma^2/4$. Thus, $(\mathbf{w}^* - \mathbf{w})^T \mathbf{M}^{\mathcal{D}'}(\mathbf{w}^* - \mathbf{w}) = \mathbf{E}_{(\mathbf{x},y) \sim \mathcal{D}'}[(\langle \mathbf{w}^* - \mathbf{w}, \mathbf{x} \rangle)^2] \geq \delta \gamma^2/8$, completing the proof. $\qquad\square$

Claim 2.2 says that $\mathbf{w}^* - \mathbf{w}$ has a large component on the large eigenvalues of $\mathbf{M}^{\mathcal{D}'}$. Building on this claim, we obtain the following result:

**Lemma 2.3.** *Let $\mathbf{w}^*, \mathbf{w}, \mathbf{M}^{\mathcal{D}'}$ be as in Claim 2.2. There exists $k \in \mathbb{Z}_+$ so that if $V_k$ is the span of the top $k$ eigenvectors of $\mathbf{M}^{\mathcal{D}'}$, we have that $\|\mathrm{Proj}_{V_k}(\mathbf{w}^* - \mathbf{w})\|_2^2 \geq k\delta\gamma^2/8$.*

*Proof.* Note that the matrix $\mathbf{M}^{\mathcal{D}'}$ is PSD and let $0 > \lambda_{\max} = \lambda_1 \geq \lambda_2 \geq \ldots \geq \lambda_d \geq 0$ be its set of eigenvalues. We will denote by $V_{\geq t}$ the space spanned by the eigenvectors of $\mathbf{M}^{\mathcal{D}'}$ corresponding to eigenvalues of magnitude at least $t$. Let $d_t = \dim(V_{\geq t})$ be the dimension of $V_{\geq t}$, i.e., the number of $i \in [d]$ with $\lambda_i \geq t$. Since $\mathbf{x}$ is supported on the unit ball, for $(\mathbf{x}, y) \sim \mathcal{D}'$, we have that $\mathrm{tr}(\mathbf{M}^{\mathcal{D}'}) = \mathbf{E}_{(\mathbf{x},y) \sim \mathcal{D}'}[\mathrm{tr}(\mathbf{x}\mathbf{x}^T)] \leq 1$. Since $\mathbf{M}^{\mathcal{D}'}$ is PSD, we have that $\mathrm{tr}(\mathbf{M}^{\mathcal{D}'}) = \sum_{i=1}^d \lambda_i$ and we can write

$$1 \geq \mathrm{tr}(\mathbf{M}^{\mathcal{D}'}) = \sum_{i=1}^d \lambda_i = \sum_{i=1}^d \int_0^{\lambda_i} 1 dt = \sum_{i=1}^d \int_0^{\lambda_{\max}} \mathbf{1}_{\lambda_i \geq t} dt = \int_0^{\lambda_{\max}} d_t dt, \tag{1}$$

where the last equality follows by changing the order of the summation and the integration. If the projection of $(\mathbf{w}^* - \mathbf{w})$ onto the $i$-th eigenvector of $\mathbf{M}^{\mathcal{D}'}$ has $\ell_2$-norm $a_i$, we have that

$$\delta\gamma^2/8 \leq (\mathbf{w}^* - \mathbf{w})^T \mathbf{M}^{\mathcal{D}'}(\mathbf{w}^* - \mathbf{w}) = \sum_{i=1}^d \lambda_i a_i^2 = \sum_{i=1}^d \int_0^{\lambda_{\max}} a_i^2 \mathbf{1}_{\lambda_i \geq t} dt = \int_0^{\lambda_{\max}} \|\mathrm{Proj}_{V_{\geq t}}(\mathbf{w}^* - \mathbf{w})\|_2^2 dt, \tag{2}$$

where the first inequality uses Claim 2.2, the first equality follows by the Pythagorean theorem, and the last equality follows by changing the order of the summation and the integration. Combining (1) and (2), we obtain $\int_0^{\lambda_{\max}} \|\mathrm{Proj}_{V_{\geq t}}(\mathbf{w}^* - \mathbf{w})\|_2^2 dt \geq (\delta\gamma^2/8) \int_0^{\lambda_{\max}} d_t dt$. By an averaging argument, there exists $0 \leq t \leq \lambda_{\max}$ such that $\|\mathrm{Proj}_{V_{\geq t}}(\mathbf{w}^* - \mathbf{w})\|_2^2 \geq (\delta\gamma^2/8)d_t$. Letting $k = d_t$ and noting that $V_{\geq t} = V_k$ completes the proof. $\qquad\square$

Lemma 2.3 suggests a method for producing an approximation to $\mathbf{w}^*$, or more precisely a vector that produces empirical $\gamma/2$-margin error at most $(1 + \delta)\mathrm{OPT}_\gamma^{\mathcal{D}}$. We start by describing a non-deterministic procedure, which we will then turn into an actual algorithm.

The method proceeds in a sequence of stages. At stage $i$, we have a hypothesis weight vector $\mathbf{w}^{(i)}$. (At stage $i = 0$, we start with $\mathbf{w}^{(0)} = \mathbf{0}$.) At any stage $i$, if $\mathrm{err}_{\gamma/2}^{\mathcal{D}}(\mathbf{w}^{(i)}) \le (1 + \delta)\mathrm{OPT}_\gamma^{\mathcal{D}}$, then $\mathbf{w}^{(i)}$ is a sufficient estimator. Otherwise, we consider the matrix $\mathbf{M}^{(i)} = \mathbf{E}_{(\mathbf{x},y) \sim \mathcal{D}^{(i)}}[\mathbf{x}\mathbf{x}^T]$, where $\mathcal{D}^{(i)}$ is $\mathcal{D}$ conditioned on $y\langle \mathbf{w}^{(i)}, \mathbf{x}\rangle \le \gamma/2$. By Lemma 2.3, we know that for some positive integer value $k^{(i)}$, we have that the projection of $\mathbf{w}^* - \mathbf{w}^{(i)}$ onto $V_{k^{(i)}}$ has squared norm at least $\delta k^{(i)}\gamma^2/8$.

Let $\mathbf{p}^{(i)}$ be this projection. We set $\mathbf{w}^{(i+1)} = \mathbf{w}^{(i)} + \mathbf{p}^{(i)}$. Since the projection of $\mathbf{w}^* - \mathbf{w}^{(i)}$ and its complement are orthogonal, we have

$$\|\mathbf{w}^* - \mathbf{w}^{(i+1)}\|_2^2 = \|\mathbf{w}^* - \mathbf{w}^{(i)}\|_2^2 - \|\mathbf{p}^{(i)}\|_2^2 \le \|\mathbf{w}^* - \mathbf{w}^{(i)}\|_2^2 - \delta k^{(i)}\gamma^2/8 , \qquad (3)$$

where the inequality uses the fact that $\|\mathbf{p}^{(i)}\|_2^2 \ge k^{(i)}\delta\gamma^2/8$ (as follows from Lemma 2.3). Let $s$ be the total number of stages. We can write

$$1 \ge \|\mathbf{w}^* - \mathbf{w}^{(0)}\|_2^2 - \|\mathbf{w}^* - \mathbf{w}^{(s)}\|_2^2 = \sum_{i=0}^{s-1}\left(\|\mathbf{w}^* - \mathbf{w}^{(i)}\|_2^2 - \|\mathbf{w}^* - \mathbf{w}^{(i+1)}\|_2^2\right) \ge (\delta\gamma^2/8)\sum_{i=0}^{s-1} k^{(i)} ,$$

where the first inequality uses that $\|\mathbf{w}^* - \mathbf{w}^{(0)}\|_2^2 = 1$ and $\|\mathbf{w}^* - \mathbf{w}^{(s)}\|_2^2 \ge 0$, the second notes the telescoping sum, and the third uses (3). Therefore, $s \le \sum_{i=0}^{s-1} k^{(i)} \le 8/(\delta\gamma^2)$. Therefore, the above procedure terminates after at most $8/(\delta\gamma^2)$ stages at some $\mathbf{w}^{(s)}$ with $\mathrm{err}_{\gamma/2}^{\mathcal{D}}(\mathbf{w}^{(s)}) \le (1 + \delta)\mathrm{OPT}_\gamma^{\mathcal{D}}$.

We now describe how to turn the above procedure into an actual algorithm. Our algorithm tries to simulate the above described procedure by making appropriate guesses. In particular, we start by guessing a sequence of positive integers $k^{(i)}$ whose sum is at most $8/(\delta\gamma^2)$. This can be done in $2^{O(1/(\delta\gamma^2))}$ ways. Next, given this sequence, our algorithm guesses the vectors $\mathbf{w}^{(i)}$ over all $s$ stages in order. In particular, given $\mathbf{w}^{(i)}$, the algorithm computes the matrix $\mathbf{M}^{(i)}$ and the subspace $V_{k^{(i)}}$, and guesses the projection $\mathbf{p}^{(i)} \in V_{k^{(i)}}$, which then gives $\mathbf{w}^{(i+1)}$. Of course, we cannot expect our algorithm to guess $\mathbf{p}^{(i)}$ exactly (as there are infinitely many points in $V_{k^{(i)}}$), but we can guess it to within $\ell_2$-error $\mathrm{poly}(\gamma)$, by taking an appropriate net. This involves an additional guess of size $(1/\gamma)^{O(k^{(i)})}$ in each stage. In total, our algorithm makes $2^{\tilde{O}(1/(\delta\gamma^2))}$ many different guesses.

We note that the sample version of our algorithm is essentially identical to the idealized version described above, by replacing the distribution $\mathcal{D}$ by its empirical version and leveraging the following statistical bound:

**Fact 2.4** ([BM02, McA03]). *Let $S = \{(\mathbf{x}^{(i)}, y^{(i)})\}_{i=1}^m$ be a multiset of i.i.d. samples from $\mathcal{D}$, where $m = \Omega(\log(1/\tau)/(\epsilon^2\gamma^2))$, and $\widehat{\mathcal{D}}_m$ be the empirical distribution on $S$. Then with probability at least $1 - \tau$ over $S$, simultaneously for all unit vectors $\mathbf{w}$ and margins $\gamma > 0$, if $h_{\mathbf{w}}(\mathbf{x}) = \mathrm{sign}(\langle \mathbf{w}, \mathbf{x}\rangle)$, we have that $\mathrm{err}_{0-1}^{\mathcal{D}}(h_{\mathbf{w}}) \le \mathrm{err}_\gamma^{\widehat{\mathcal{D}}_m}(\mathbf{w}) + \epsilon$.*

The pseudo-code of our algorithm is given below in Algorithm 1.

To show the correctness of the algorithm, we begin by noting that the set $C$ of candidate weight vectors produced has size $2^{\tilde{O}(1/(\delta\gamma^2))}$. This is because there are only $2^{O(1/(\delta\gamma^2))}$ many possibilities for the sequence of $k^{(i)}$, and for each such sequence the product of the sizes of the $C^{(i)}$ is $(1/(\delta\gamma))^{O(\sum k^{(i)})} = 2^{\tilde{O}(1/(\delta\gamma^2))}$. We note that, by the aforementioned analysis, for any choice of $k^{(0)}, \dots, k^{(i-1)}$ and $\mathbf{w}^{(i)}$, we either have that $\mathrm{err}_{\gamma/2}^{\widehat{\mathcal{D}}_m}(\mathbf{w}^{(i)}) \le (1 + \delta)\mathrm{OPT}_\gamma^{\widehat{\mathcal{D}}_m}$ or there is a choice of $k^{(i)}$ and $\mathbf{p}^{(i)} \in C^{(i)}$ such that

$$\|\mathbf{w}^* - \mathbf{w}^{(i)} - \mathbf{p}^{(i)}\|_2^2 \le \|\mathbf{w}^* - \mathbf{w}^{(i)}\|_2^2 - \delta k^{(i)}\gamma^2/8 + O(\delta^2\gamma^6) ,$$

where we used (3) and the fact that $C^{(i)}$ is a $\delta\gamma^3$-cover of $V_{k^{(i)}}$. Following the execution path of the algorithm, we either find some $\mathbf{w}^{(i)}$ with $\mathrm{err}_{\gamma/2}^{\widehat{\mathcal{D}}_m}(\mathbf{w}^{(i)}) \le (1 + \delta)\mathrm{OPT}_\gamma^{\widehat{\mathcal{D}}_m}$, or we find a $\mathbf{w}^{(i)}$ with

$$\|\mathbf{w}^* - \mathbf{w}^{(i)}\|_2^2 \le 1 - \left(\sum_{j=0}^{i-1} k^{(j)}\right)\delta\gamma^2/8 + O(\delta\gamma^4) ,$$

---

**Algorithm 1** Near-Optimal $(1+\delta)$-Agnostic Proper Learner

---

1: Draw a multiset $S = \{(\mathbf{x}^{(i)}, y^{(i)})\}_{i=1}^{m}$ of i.i.d. samples from $\mathcal{D}$, where $m = \Omega(\log(1/\tau)/(\epsilon^2 \gamma^2))$.
2: Let $\widehat{\mathcal{D}}_m$ be the empirical distribution on $S$.
3: **for** all sequences $k^{(0)}, k^{(1)}, \ldots, k^{(s-1)}$ of positive integers with sum at most $8/(\delta \gamma^2) + 2$ **do**
4:      Let $\mathbf{w}^{(0)} = \mathbf{0}$.
5:      **for** $i = 0, 1, \ldots, s-1$ **do**
6:          Let $\mathcal{D}^{(i)}$ be $\widehat{\mathcal{D}}_m$ conditioned on $y\langle \mathbf{w}^{(i)}, \mathbf{x} \rangle \leq \gamma/2$.
7:          Let $\mathbf{M}^{(i)} = \mathbf{E}_{(\mathbf{x}, y) \sim \mathcal{D}^{(i)}}[\mathbf{x}\mathbf{x}^T]$.
8:          Use SVD on $\mathbf{M}^{(i)}$ to find a basis for $V_{k^{(i)}}$, the span of the top $k^{(i)}$ eigenvectors.
9:          Let $C^{(i)}$ be a $\delta\gamma^3$-cover, in $\ell_2$-norm, of $V_{k^{(i)}} \cap \mathbb{B}_d$ of size $(1/(\delta\gamma))^{O(k^{(i)})}$.
10:          For each $\mathbf{p}^{(i)} \in C^{(i)}$ repeat the next step of the for loop with $\mathbf{w}^{(i+1)} = \mathbf{w}^{(i)} + \mathbf{p}^{(i)}$.
11:      **end for**
12: **end for**
13: Let $C$ denote the set of all $\mathbf{w}^{(i)}$ generated in the above loop.
14: Let $\mathbf{v} \in \operatorname{argmin}_{\mathbf{w} \in C} \operatorname{err}_{\gamma/2}^{\widehat{\mathcal{D}}_m}(\mathbf{w})$.
15: **return** Output the hypothesis $h_{\mathbf{v}}(\mathbf{x}) = \operatorname{sign}(\langle \mathbf{v}, \mathbf{x} \rangle)$.

---

where the last term is an upper bound for $\left(\sum_{j=0}^{i-1} k^{(j)}\right) \cdot O(\delta^2 \gamma^6)$. Note that this sequence terminates in at most $O(1/(\delta\gamma^2))$ stages, when it becomes impossible that $\sum k^{(j)} > 8/(\delta\gamma^2) + 1$. Thus, the output of our algorithm must contain some weight vector $\mathbf{v}$ with $\operatorname{err}_{\gamma/2}^{\widehat{\mathcal{D}}_m}(\mathbf{v}) \leq (1+\delta)\operatorname{OPT}_{\gamma}^{\widehat{\mathcal{D}}_m}$. The proof now follows by an application of Fact 2.4. This completes the proof of Theorem 2.1.

## 3   Computational Hardness Results

In this section, we provide several computational lower bounds for agnostic learning of halfspaces with a margin. To clarify the statements below, we note that we say "there is no algorithm that runs in time $T(d, \frac{1}{\gamma}, \frac{1}{\varepsilon})$" to mean that no $T(d, \frac{1}{\gamma}, \frac{1}{\varepsilon})$-time algorithm works for *all* combinations of parameters $d, \gamma$ and $\varepsilon$. (Note that we discuss the lower bounds with stronger quatifiers in the full version [DKM19].) Moreover, we also ignore the dependency on $\tau$ (the probability that the learner can be incorrect), since we only use a fixed $\tau$ (say $1/3$) in all the bounds below.

First, we show that, for any constant $\alpha > 1$, $\alpha$-agnostic learning of $\gamma$-margin halfspaces requires $2^{(1/\gamma)^{2-o(1)}}\operatorname{poly}(d, 1/\varepsilon)$ time. Up to the lower order term $\gamma^{o(1)}$ in the exponent, this matches with our algorithm (in Theorem 2.1). In fact, we show an even stronger result, that if the dependency of the running time on the margin is say $2^{(1/\gamma)^{1.99}}$, then one has to pay $2^{d^{1-o(1)}}$ in the running time.

This result holds assuming the so-called (randomized) exponential time hypothesis (ETH) [IP01, IPZ01], which postulates that there is no (randomized) algorithm that can solve 3SAT in time $2^{o(n)}$, where $n$ denotes the number of variables. ETH is a standard hypothesis used in proving (tight) running time lower bounds. We do not discuss ETH further here, but interested readers may refer to a survey by Lokshtanov *et al.* [LMS11] for an in-depth discussion and several applications of ETH.

Our first lower bound can be stated more precisely as follows:

**Theorem 3.1.** *Assuming the (randomized) ETH, for any universal constant $\alpha \geq 1$, there is no proper $\alpha$-agnostic learner for $\gamma$-margin halfspaces that runs in time $O(2^{(1/\gamma)^{2-o(1)}} 2^{d^{1-o(1)}}) f(\frac{1}{\varepsilon})$ for any function $f$.*

Secondly, we address the question of whether we can achieve $\alpha = 1$ (standard agnostic learning) while retaining running time similar to our algorithm. We answer this in the negative (assuming a standard parameterized complexity assumption): there is no $f(\frac{1}{\gamma})\operatorname{poly}(d, \frac{1}{\varepsilon})$-time 1-agnostic learner for any function $f$ (e.g., even for $f(\frac{1}{\gamma}) = 2^{2^{2^{1/\gamma}}}$). This demonstrates a stark contrast between what we can achieve with and without approximation.

**Theorem 3.2.** *Assuming W[1] is not contained in randomized FPT, there is no proper* $1$-*agnostic learner for $\gamma$-margin halfspaces that runs in time $f(\frac{1}{\gamma})\mathrm{poly}(d, \frac{1}{\varepsilon})$ for any function $f$.*

Finally, we explore the other extreme of the trade-off between the running time and approximation ratio, by asking: what is the best approximation ratio we can achieve if we only consider proper learners that run in $\mathrm{poly}(d, \frac{1}{\varepsilon}, \frac{1}{\gamma})$-time? On this front, it is known [Ser01] that the perceptron algorithm achieves $1/\gamma$-approximation. We show that a significant improvement over this is unlikely, by showing that $(1/\gamma)^{\frac{1}{\mathrm{polyloglog}(1/\gamma)}}$-approximation is not possible unless NP = RP. If we additionally assume the so-called Sliding Scale Conjecture [BGLR94], this ratio can be improved to $(1/\gamma)^c$ for some constant $c > 0$.

**Theorem 3.3.** *Assuming NP $\neq$ RP, there is no proper $(1/\gamma)^{1/\mathrm{polyloglog}(1/\gamma)}$-agnostic learner for $\gamma$-margin halfspaces that runs in time $\mathrm{poly}(d, \frac{1}{\varepsilon}, \frac{1}{\gamma})$. Furthermore, assuming NP $\neq$ RP and the Sliding Scale Conjecture [BGLR94], there is no proper $(1/\gamma)^c$-agnostic learning for $\gamma$-margin halfspaces that runs in time $\mathrm{poly}(d, \frac{1}{\varepsilon}, \frac{1}{\gamma})$ for some constant $c > 0$.*

Due to the technical nature of the Sliding Scale Conjecture, we do not state it in full here; please refer to the full version for a formal statement [DKM19].

We note here that the constant $c$ in Theorem 3.3 is not explicit, i.e., it depends on the constant from the Sliding Scale Conjecture (SSC). Moreover, even when assuming the most optimistic parameters of SSC, the constant $c$ we can get is still very small. For instance, it is still possible that a say $\sqrt{1/\gamma}$-agnostic learning algorithm that runs in polynomial time exists, and this remains an interesting open question. We remark that Daniely *et al*. [DLS14] have made partial progress in this direction by showing that, any $\mathrm{poly}(d, \frac{1}{\varepsilon}, \frac{1}{\gamma})$-time learner that belongs to a "generalized linear family" cannot achieve approximation ratio $\alpha$ better than $\Omega\left(\frac{1/\gamma}{\mathrm{polylog}(1/\gamma)}\right)$. We note that the inapproximability ratio of [DLS14] is close to being tight for a natural, yet restricted, family of improper learners. On the other hand, our proper hardness result holds against *all* proper learners under a widely believed worst-case complexity assumption.

Due to space limitations, the proofs of our hardness results are deferred to the full version of this work [DKM19].

# 4 Conclusions and Open Problems

This work gives nearly tight upper and lower bounds for the problem of $\alpha$-agnostic proper learning of halfspaces with a margin, for $\alpha = O(1)$. Our upper and lower bounds for $\alpha = \omega(1)$ are far from tight. Closing this gap is an interesting open problem. Charactering the fine-grained complexity of the problem for improper learning algorithms remains a challenging open problem.

More broadly, an interesting direction for future work would be to generalize our agnostic learning results to broader classes of geometric functions. Finally, we believe that finding further connections between the problem of agnostic learning with a margin and adversarially robust learning is an intriguing direction to be explored.

**Acknowledgments**

Part of this work was performed while Ilias Diakonikolas was at the Simons Institute for the Theory of Computing during the program on Foundations of Data Science. Ilias Diakonikolas is supported by Supported by NSF Award CCF-1652862 (CAREER) and a Sloan Research Fellowship. Daniel M. Kane is supported by NSF Award CCF-1553288 (CAREER) and a Sloan Research Fellowship.

## Footnotes

[3]To avoid clutter in the expressions, we will henceforth assume that the failure probability $\tau = 1/10$. Recall that one can always boost the confidence probability with an $O(\log(1/\tau))$ multiplicative overhead.

[4]Here the constant 0.99 can be replaced by any constant less than one, with an appropriate increase to the algorithm's running time.

[5]We use $\text{OPT}_{0-1}^{\mathcal{D}} \overset{\text{def}}{=} \min_{\mathbf{w}\in\mathbb{R}^d} \text{err}_{0-1}^{\mathcal{D}}(\mathbf{w})$ to denote the minimum error rate achievable by any halfspace.

[6]In other words, for any assignment to the variables, at least 0.1 fraction of the clauses are unsatisfied.

[7]ETH states that the *exact* version of 3SAT cannot be solved in $2^{o(n)}$ time.

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
