[Supplementary Material · LTF-margin-supp.pdf]

**Supplementary Material**

 # A Omitted Results from Section 2

 ## A.1 Warm-Up: Basic Algorithm

385 In this subsection, we present a basic algorithm that achieves $\alpha = 1$ and whose runtime is
386 $\text{poly}(d)2^{\tilde{O}(1/(\epsilon\gamma^2))}$. Despite its slow runtime, this algorithm serves as a warm-up for our more
387 sophisticated constant factor approximation algorithm in the next subsection.

388 We start by establishing a basic structural property of this setting which motivates our basic algorithm.
389 We start with the following simple claim:

390 **Claim A.1.** *Let* $\mathbf{M}^{\mathcal{D}} = \mathbf{E}_{(\mathbf{x},y)\sim\mathcal{D}}[\mathbf{x}\mathbf{x}^T]$ *and* $\mathbf{w}^*$ *be a unit vector such that* $\text{err}_\gamma^{\mathcal{D}}(\mathbf{w}^*) \leq \text{OPT}_\gamma^{\mathcal{D}} \leq$
391 $1/2$. *Then,* $\|\mathbf{M}^{\mathcal{D}}\|_2 \geq \mathbf{w}^{*T}\mathbf{M}^{\mathcal{D}}\mathbf{w}^* \geq \gamma^2/2$.

392 *Proof.* By assumption, $\mathbf{Pr}_{(\mathbf{x},y)\sim\mathcal{D}}[|\langle\mathbf{w}^*,\mathbf{x}\rangle| \geq \gamma] \geq 1/2$, which implies that
393 $\mathbf{E}_{(\mathbf{x},y)\sim\mathcal{D}}[(\langle\mathbf{w}^*,\mathbf{x}\rangle)^2] \geq \gamma^2/2$. The claim follows from the fact that $\mathbf{v}^T\mathbf{M}^{\mathcal{D}}\mathbf{v} =$
394 $\mathbf{E}_{(\mathbf{x},y)\sim\mathcal{D}}[(\langle\mathbf{v},\mathbf{x}\rangle)^2]$, for any $\mathbf{v} \in \mathbb{R}^d$, and the definition of the spectral norm. $\qquad\square$

395 Claim A.1 allows us to obtain an approximation to the optimal halfspace by projecting on the space
396 of large eigenvalues of $\mathbf{M}^{\mathcal{D}}$. We will need the following terminology: For $\delta > 0$, let $V_{\geq\delta}^{\mathcal{D}}$ be the
397 space spanned by the eigenvalues of $\mathbf{M}^{\mathcal{D}}$ with magnitude at least $\delta$ and $V_{<\delta}^{\mathcal{D}}$ be its complement. Let
398 $\text{Proj}_V(\mathbf{v})$ denote the projection operator of vector $\mathbf{v}$ on subspace $V$. Then, we have the following:

399 **Lemma A.2.** *Let* $\delta > 0$ *and* $\mathbf{w}' = \text{Proj}_{V_{\geq\delta}^{\mathcal{D}}}(\mathbf{w}^*)$. *Then,* $\text{err}_{\gamma/2}^{\mathcal{D}}(\mathbf{w}') \leq \text{err}_\gamma^{\mathcal{D}}(\mathbf{w}^*) + 4\delta/\gamma^2$.

400 *Proof.* Let $\mathbf{w}^* = \mathbf{w}' + \mathbf{w}''$, where $\mathbf{w}'' = \text{Proj}_{V_{<\delta}^{\mathcal{D}}}(\mathbf{w}^*)$. Observe that for any $(\mathbf{x},y)$, if
401 $y\langle\mathbf{w}',\mathbf{x}\rangle \leq \gamma/2$ then $y\langle\mathbf{w}^*,\mathbf{x}\rangle \leq \gamma$, unless $|\langle\mathbf{w}'',\mathbf{x}\rangle| \geq \gamma/2$. Hence, $\text{err}_{\gamma/2}^{\mathcal{D}}(\mathbf{w}') \leq \text{err}_\gamma^{\mathcal{D}}(\mathbf{w}^*) +$
402 $\mathbf{Pr}_{(\mathbf{x},y)\sim\mathcal{D}}[|\langle\mathbf{w}'',\mathbf{x}\rangle| \geq \gamma/2]$. By definition of $\mathbf{w}''$ and $\mathbf{M}^{\mathcal{D}}$, we have that $\mathbf{E}_{(\mathbf{x},y)\sim\mathcal{D}}[(\langle\mathbf{w}'',\mathbf{x}\rangle)^2] \leq \delta$.
403 By Markov's inequality, we thus obtain $\mathbf{Pr}_{(\mathbf{x},y)\sim\mathcal{D}}[(\langle\mathbf{w}'',\mathbf{x}\rangle)^2 \geq \gamma^2/4] \leq 4\delta/\gamma^2$, completing the
404 proof of the lemma. $\qquad\square$

405 Motivated by Lemma A.2, the idea is to enumerate over $V_{\geq\delta}^{\mathcal{D}}$, for $\delta = \Theta(\epsilon\gamma^2)$, and output a vector $\mathbf{v}$
406 with smallest empirical $\gamma/2$-margin error. To turn this into an actual algorithm, we work with a finite
sample set and enumerate over an appropriate cover of the space $V_{\geq\delta}^{\mathcal{D}}$. The pseudocode is as follows:

---

**Algorithm 2** Basic 1-Agnostic Proper Learning Algorithm

1: Draw a multiset $S = \{(\mathbf{x}^{(i)}, y^{(i)})\}$ of i.i.d. samples from $\mathcal{D}$, where $m = \Omega(\log(1/\tau)/(\epsilon^2\gamma^2))$.
2: Let $\widehat{\mathcal{D}}_m$ be the empirical distribution on $S$.
3: Let $\mathbf{M}^{\widehat{\mathcal{D}}_m} = \mathbf{E}_{(\mathbf{x},y)\sim\widehat{\mathcal{D}}_m}[\mathbf{x}\mathbf{x}^T]$.
4: Set $\delta = \epsilon\gamma^2/16$. Use SVD to find a basis of $V_{\geq\delta}^{\widehat{\mathcal{D}}_m}$.
5: Compute a $\delta/2$-cover, $C_{\delta/2}$, in $\ell_2$-norm, of $V_{\geq\delta}^{\widehat{\mathcal{D}}_m} \cap \mathbb{S}_{d-1}$.
6: Let $\mathbf{v} \in \text{argmin}_{\mathbf{w}\in C_{\delta/2}} \text{err}_{\gamma/4}^{\widehat{\mathcal{D}}_m}(\mathbf{w})$.
7: **return** $h_{\mathbf{v}}(\mathbf{x}) = \text{sign}(\langle\mathbf{v},\mathbf{x}\rangle)$.

---

407

408 First, we analyze the runtime of our algorithm. The SVD of $\mathbf{M}^{\widehat{\mathcal{D}}_m}$ can be computed in $\text{poly}(d/\delta)$
409 time. Note that $V_{\geq\delta}^{\widehat{\mathcal{D}}_m}$ has dimension at most $1/\delta$. This follows from the fact that $\mathbf{M}^{\widehat{\mathcal{D}}_m}$ is PSD
410 and its trace is $\sum_{i=1}^d \lambda_i = \text{tr}(\mathbf{M}^{\widehat{\mathcal{D}}_m}) = \mathbf{E}_{(\mathbf{x},y)\sim\widehat{\mathcal{D}}_m}[\text{tr}(\mathbf{x}\mathbf{x}^T)] \leq 1$, where we used that $\|\mathbf{x}\|_2 \leq 1$
411 with probability 1 over $\widehat{\mathcal{D}}_m$. Therefore, the unit sphere of $V_{\geq\delta}^{\widehat{\mathcal{D}}_m}$ has a $\delta/2$-cover $C_{\delta/2}$ of size
412 $(2/\delta)^{O(1/\delta)} = 2^{\tilde{O}(1/(\epsilon\gamma^2))}$ that can be computed in output polynomial time.

413 We now prove correctness. The main idea is to apply Lemma A.2 for the empirical distribution
414 $\widehat{\mathcal{D}}_m$ combined with Fact 2.4. We proceed with the formal proof: First, we claim that for $m = $
415 $\Omega(\log(1/\tau)/\epsilon^2)$, with probability at least $1-\tau/2$ over $S$, we have that $\mathrm{err}_\gamma^{\widehat{\mathcal{D}}_m}(\mathbf{w}^*) \leq \mathrm{err}_\gamma^{\mathcal{D}}(\mathbf{w}^*)+\epsilon/8$.
416 To see this, note that $\mathrm{err}_\gamma^{\widehat{\mathcal{D}}_m}(\mathbf{w}^*)$ can be viewed as a sum of Bernoulli random variables with
417 expectation $\mathrm{err}_\gamma^{\mathcal{D}}(\mathbf{w}^*)$. Hence, the claim follows by a Chernoff bound. By an argument similar to
418 that of Lemma A.2, we have that $\mathrm{err}_{\gamma/4}^{\widehat{D}_m}(\mathbf{v}) \leq \mathrm{err}_{\gamma/2}^{\widehat{D}_m}(\mathbf{w}') + \epsilon/2$. Indeed, we can write $\mathbf{v} = \mathbf{w}' + \mathbf{r}$,
419 where $\|\mathbf{r}\|_2 \leq \delta/2$, and follow the same argument.

420 In summary, we have the following sequence of inequalities:

$$
\begin{aligned}
\mathrm{err}_{\gamma/4}^{\widehat{D}_m}(\mathbf{v}) &\leq \mathrm{err}_{\gamma/2}^{\widehat{D}_m}(\mathbf{w}') + \epsilon/2 \leq \mathrm{err}_\gamma^{\widehat{D}_m}(\mathbf{w}^*) + \epsilon/2 + \epsilon/4 \\
&\leq \mathrm{err}_\gamma^{\mathcal{D}}(\mathbf{w}^*) + \epsilon/2 + \epsilon/4 + \epsilon/8 \,,
\end{aligned}
$$

421 where the second inequality uses Lemma A.2 for $\widehat{\mathcal{D}}_m$. Finally, we use Fact 2.4 for $\gamma/4$ and $\epsilon/8$ to
422 obtain that $\mathrm{err}_{0-1}^{\mathcal{D}}(h_\mathbf{v}) \leq \mathrm{err}_{\gamma/4}^{\widehat{D}_m}(\mathbf{v}) + \epsilon/8 \leq \mathrm{OPT}_\gamma^{\mathcal{D}} + \epsilon$. The proof follows by a union bound.

## A.2 Algorithm for $\alpha$-Agnostic Learning

424 In this section, we show that if one wishes to obtain an $\alpha$-agnostic learner for some large $\alpha \gg 1$, one
425 can obtain runtime exponential in $1/(\alpha\gamma)^2$ rather than $1/\gamma^2$. Formally we prove:

**Theorem A.3.** *There is an algorithm that uses $\tilde{O}(1/(\epsilon^2\gamma^2))$ samples, runs in time $\mathrm{poly}(d) \cdot$*
427 *$(1/\epsilon)^{\tilde{O}(1/(\alpha\gamma)^2)}$ and is a $\alpha$-agnostic proper learner for $\gamma$-margin halfspaces with probability $9/10$.*

428 Let $\mathcal{D}$ be a distribution over $\mathbb{B}^d \times \{-1, 1\}$. Suppose that there exists a unit vector $\mathbf{w}^* \in \mathbb{R}^d$ such that
429 $\mathbf{Pr}_{(\mathbf{x},y)\sim\mathcal{D}}[y\langle\mathbf{w}^*, \mathbf{x}\rangle \geq \gamma] \geq 1 - \mathrm{OPT}_\gamma^{\mathcal{D}}$ for some $\mathrm{OPT}_\gamma^{\mathcal{D}} > 0$. Suppose additionally that $\gamma, \epsilon > 0$
430 and $\alpha > 1$. We will give an algorithm that given sample access to $\mathcal{D}$ along with $\gamma, \alpha, \epsilon, \mathrm{OPT}_\gamma^{\mathcal{D}}$,
431 draws $O(\log(\alpha/\epsilon)/(\gamma\epsilon)^2)$ samples, runs in time $\mathrm{poly}(d)(1/\gamma\epsilon)^{\tilde{O}(1/(\alpha\gamma)^2)}$ and with probability at
432 least $9/10$ returns a $\mathbf{w}$ with $\mathbf{Pr}_{(\mathbf{x},y)\sim\mathcal{D}}[\mathrm{sign}(\langle\mathbf{w}, \mathbf{x}\rangle) \neq y] < O(\alpha \cdot \mathrm{OPT}_\gamma^{\mathcal{D}} + \epsilon)$.

433 We begin by providing an algorithm that works if the distribution $\mathcal{D}$ is given explicitly. We will be
434 able to reduce to this case by using the empirical distribution over a sufficiently large set of samples.

**Proposition A.4.** *Let $\mathcal{D}$ be an explicit distribution over $\mathbb{B}^d \times \{-1, 1\}$. Suppose that there exists*
436 *a unit vector $\mathbf{w}^*$ so that $\mathbf{Pr}_{(\mathbf{x},y)\sim\mathcal{D}}[y\langle\mathbf{w}^*, \mathbf{x}\rangle \geq \gamma] \geq 1 - \mathrm{OPT}_\gamma^{\mathcal{D}}$ for some $\mathrm{OPT}_\gamma^{\mathcal{D}} > 0$. Sup-*
437 *pose additionally that $\gamma, \epsilon > 0$ and $\alpha > 1$. There exists an algorithm that given $\mathcal{D}$ along with*
438 *$\gamma, \alpha, \epsilon, \mathrm{OPT}_\gamma^{\mathcal{D}}$, runs in time $\mathrm{poly}(d)(|\mathrm{supp}(\mathcal{D})|/(\alpha\gamma \cdot \mathrm{OPT}_\gamma^{\mathcal{D}}))^{\tilde{O}(1/(\alpha\gamma)^2)}$ and returns a $\mathbf{w}$ with*
439 *$\mathbf{Pr}_{(\mathbf{x},y)\sim\mathcal{D}}[\mathrm{sign}(\langle\mathbf{w}, \mathbf{x}\rangle) \neq y] < O(\alpha \cdot \mathrm{OPT}_\gamma^{\mathcal{D}} + \epsilon)$.*

440 Our main technical tool here will be the vector of Chow parameters [Cho61, OS08, DDFS14]:

**Definition A.5.** *Given a boolean function $f : \mathbb{B}^d \to \{\pm 1\}$, and a distribution $\mathcal{D}$ on $\mathbb{B}^d$ the Chow*
442 *parameters vector of $f$, is the vector $\mathbf{Chow}(f)$ given by the expectation $\mathbf{E}_{\mathbf{x}\sim\mathcal{D}}[f(\mathbf{x})\mathbf{x}]$.*

443 We note that learning the Chow parameters of the halfspace $f_{\mathbf{w}^*}(x) = \mathrm{sign}(\langle\mathbf{w}^*, \mathbf{x}\rangle)$ determines the
444 function $f_{\mathbf{w}^*}$ up to small error. In particular:

**Lemma A.6.** *Let $g : \mathbb{B}^d \to \{\pm 1\}$ satisfy $\mathbf{Pr}_{\mathbf{x}\sim\mathcal{D}}[f_{\mathbf{w}^*}(\mathbf{x}) \neq g(\mathbf{x})] \geq \epsilon$. Then, we have that*
446 *$\|\mathbf{Chow}(f_{\mathbf{w}^*}) - \mathbf{Chow}(g)\|_2 \geq \epsilon\gamma$.*

447 *Proof.* We note that

$$
\begin{aligned}
\|\mathbf{Chow}(f_{\mathbf{w}^*}) - \mathbf{Chow}(g)\|_2 &\geq \langle\mathbf{w}^*, \mathbf{Chow}(f_{\mathbf{w}^*}) - \mathbf{Chow}(g)\rangle \\
&= \mathbf{E}_{\mathbf{x}\sim D}[\langle\mathbf{w}^*, \mathbf{x}\rangle(f_{\mathbf{w}^*}(\mathbf{x}) - g(\mathbf{x}))] \\
&= \mathbf{E}_{\mathbf{x}\sim D}[|\langle\mathbf{w}^*, \mathbf{x}\rangle| - g(\mathbf{x})\langle\mathbf{v}, \mathbf{x}\rangle] \\
&= 2\mathbf{E}_{\mathbf{x}\sim D}[|\langle\mathbf{w}^*, \mathbf{x}\rangle| \cdot \mathbf{1}_{f_{\mathbf{w}^*}(\mathbf{x})\neq g(\mathbf{x})}].
\end{aligned}
$$

448 However, there is at least an $\epsilon$ probability that $f(\mathbf{x}) \neq g(\mathbf{x})$ and $y\langle\mathbf{w}^*, \mathbf{x}\rangle \geq \gamma$, which implies that
449 $|\langle\mathbf{w}^*, \mathbf{x}\rangle| \geq \gamma$. Therefore, this expectation is at least $\epsilon\gamma$. $\qquad\square$

450 Lemma A.6, combined with the gradient descent type algorithm in [DDFS14], implies that learning
451 an approximation to $\mathbf{Chow}(f_{\mathbf{w}^*})$ is sufficient to learn a good hypothesis.

452 **Lemma A.7** ([DDFS14]). *There is a polynomial time algorithm that given an explicit distribution*
453 $\mathcal{D}$ *and a vector* $\mathbf{c}$ *with* $\|\mathbf{Chow}(f_{\mathbf{w}^*}) - \mathbf{c}\|_2 \leq \epsilon\gamma$, *returns a* $\mathbf{w}$ *so that* $\mathbf{Pr}_{(\mathbf{x},y)\sim\mathcal{D}}[\mathrm{sign}(\langle\mathbf{w},\mathbf{x}\rangle) \neq$
454 $\mathrm{sign}(\langle\mathbf{w}^*,\mathbf{x}\rangle)] \leq \epsilon + \mathrm{OPT}_\gamma^{\mathcal{D}}$. *In particular, for this* $\mathbf{w}$ *we have that* $\mathbf{Pr}_{(\mathbf{x},y)\sim\mathcal{D}}[\mathrm{sign}(\langle\mathbf{w},\mathbf{x}\rangle) \neq y] =$
455 $O(\epsilon + \mathrm{OPT}_\gamma^{\mathcal{D}})$.

456 Thus, it will suffice to approximate the Chow parameters of $f_{\mathbf{w}^*}$ to error $\alpha\gamma \cdot \mathrm{OPT}_\gamma^{\mathcal{D}}$. One might
457 consider using the empirical Chow parameters, namely $P = \mathbf{E}_{(\mathbf{x},y)\sim\mathcal{D}}[y\mathbf{x}]$ for this purpose. In the
458 realizable case, this would be the right thing to do, but we are working in the agnostic setting. We
459 note that since $y = \mathrm{sign}(\langle\mathbf{w}^*,\mathbf{x}\rangle)$ for all but an $\mathrm{OPT}_\gamma^{\mathcal{D}}$-fraction of $\mathbf{x}$, and since the $\mathbf{x}$ are vectors
460 in the unit ball, the error has $\ell_2$-norm at most $\mathrm{OPT}_\gamma^{\mathcal{D}}$. In fact, if we have some vector $\mathbf{w}$ so that
461 $\langle\mathbf{w}, P - \mathbf{Chow}(f_{\mathbf{w}^*})\rangle \geq \alpha\gamma \cdot \mathrm{OPT}_\gamma^{\mathcal{D}}$, then there must be some $(\mathbf{x},y)$ in the domain of $\mathcal{D}$ with
462 $|\langle\mathbf{x},\mathbf{w}\rangle| \geq \alpha\gamma$. The idea is to guess this $\mathbf{w}$ and then guess the true projection of $\mathbf{Chow}(f_{\mathbf{w}^*})$ onto
463 $\mathbf{w}$.

We present the pseudo-code for the algorithm establishing Proposition A.4 as Algorithm 3 below.

---

**Algorithm 3** $\alpha$-Agnostic Proper Learner of Proposition A.4

---

1: Let $m$ be a sufficiently large multiple of $\log(1/\alpha\gamma)/(\alpha\gamma)^2$.
2: Let $P = \mathbf{E}_{(\mathbf{x},y)\sim\mathcal{D}}[y\mathbf{x}]$
3: **for** every sequence $\mathbf{x}^{(1)}, \ldots, \mathbf{x}^{(m)}$ from $\mathcal{D}$ **do**
4:      Let $V$ be the span of $\mathbf{x}^{(1)}, \ldots, \mathbf{x}^{(m)}$.
5:      Let $\mathcal{C}$ be a $(\alpha\gamma \cdot \mathrm{OPT}_\gamma^{\mathcal{D}})$-cover of the unit ball of $V$.
6:      **for** each $g \in \mathcal{C}$ **do**
7:          Let $P'$ be obtained by replacing the projection of $P$ onto $V$ with $g$. In particular, $P' = P - \mathrm{Proj}_V(P) + g$.
8:          Run Lemma A.7 to find a hypothesis $\mathbf{w}$.
9:      **end for**
10: **end for**
11: **return** The hypothesis that produces smallest empirical error among all $\mathbf{w}$'s in Line 8.

---

464

465 *Proof of Proposition A.4.* Firstly, note that the runtime of this algorithm is clearly
466 $\mathrm{poly}(d)\left(\frac{|\mathcal{D}|}{\mathrm{OPT}_\gamma^{\mathcal{D}}\alpha\gamma}\right)^{\tilde{O}(1/(\alpha\gamma)^2)}$. It remains to show correctness. We note that by Lemma A.7 it
467 suffices to show that some $P'$ is within $O(\alpha\gamma \cdot \mathrm{OPT}_\gamma^{\mathcal{D}})$ of $\mathbf{Chow}(f_{\mathbf{w}^*})$. For this it suffices to show
468 that there is a sequence $\mathbf{x}^{(1)}, \ldots, \mathbf{x}^{(m)}$ so that $\|\mathrm{Proj}_{V^\perp}(\mathbf{Chow}(f_{\mathbf{w}^*}) - P)\|_2 = O(\alpha\gamma\mathrm{OPT}_\gamma^{\mathcal{D}})$.

To show this, let $V_i$ be the span of $\mathbf{x}^{(1)}, \mathbf{x}^{(2)}, \ldots, \mathbf{x}^{(i)}$. We claim that if $\|\mathrm{Proj}_{V_i^\perp}(\mathbf{Chow}(f_{\mathbf{w}^*}) - P)\|_2 \gg \alpha\gamma \cdot \mathrm{OPT}_\gamma^{\mathcal{D}}$, then there exists an $\mathbf{x}^{(i+1)}$ in the support of $\mathcal{D}$ so that

$$\|\mathrm{Proj}_{V_{i+1}^\perp}(\mathbf{Chow}(f_{\mathbf{w}^*}) - P)\|_2^2 = \|\mathrm{Proj}_{V_i^\perp}(\mathbf{Chow}(f_{\mathbf{w}^*}) - P)\|_2^2(1 - \Omega(\alpha\gamma)^2).$$

To show this, we let $\mathbf{w}$ be the unit vector in the direction of $\mathrm{Proj}_{V_i^\perp}(\mathbf{Chow}(f_{\mathbf{w}^*}) - P)$. We note that

$$\|\mathrm{Proj}_{V_i^\perp}(\mathbf{Chow}(f_{\mathbf{w}^*}) - P)\|_2 = \langle\mathbf{w}, \mathbf{Chow}(f_{\mathbf{w}^*}) - P\rangle = \mathbf{E}_{(\mathbf{x},y)\sim\mathcal{D}}[\langle\mathbf{w},\mathbf{x}\rangle(\mathrm{sign}(\langle\mathbf{w}^*,\mathbf{x}\rangle) - y)] .$$

469 Since $\mathrm{sign}(\langle\mathbf{w}^*,\mathbf{x}\rangle) - y$ is 0 for all but an $\mathrm{OPT}_\gamma^{\mathcal{D}}$-fraction of $(\mathbf{x},y)$, we have that there must be
470 some $\mathbf{x}^{(i+1)}$ so that $\langle\mathbf{x}^{(i+1)},\mathbf{w}\rangle \gg \|\mathrm{Proj}_{V_i^\perp}(\mathbf{Chow}(f_{\mathbf{w}^*}) - P)\|_2/\mathrm{OPT}_\gamma^{\mathcal{D}} \gg \alpha\gamma$. If we chose this
471 $\mathbf{x}^{(i+1)}$, we have that

$$\|\mathrm{Proj}_{V_{i+1}^\perp}(\mathbf{Chow}(f_{\mathbf{w}^*}) - P)\|_2^2 \leq \|\mathrm{Proj}_{V_i^\perp}(\mathbf{Chow}(f_{\mathbf{w}^*}) - P)\|_2^2 - \langle\mathbf{x}^{(i+1)}, \mathbf{Chow}(f_{\mathbf{w}^*}) - P\rangle^2$$

$$= \|\mathrm{Proj}_{V_i^\perp}(\mathbf{Chow}(f_{\mathbf{w}^*}) - P)\|_2^2(1 - \langle\mathbf{x}^{(i+1)},\mathbf{w}\rangle^2)$$

$$= \|\mathrm{Proj}_{V_i^\perp}(\mathbf{Chow}(f_{\mathbf{w}^*}) - P)\|_2^2(1 - \Omega(\alpha\gamma)^2).$$

472  Therefore, unless some $\|\mathrm{Proj}_{V_i^\perp}(\mathbf{Chow}(f_{\mathbf{w}^*}) - P)\|_2^2 = O(\alpha\gamma \cdot \mathrm{OPT}_\gamma^{\mathcal{D}})$, we have that

$$\|\mathrm{Proj}_{V_m^\perp}(\mathbf{Chow}(f_{\mathbf{w}^*}) - P)\|_2^2 = \|P - \mathbf{Chow}(f_{\mathbf{w}^*})\|_2^2 \exp(-\Omega(m(\alpha\gamma)^2))$$
$$\leq \mathrm{OPT}_\gamma^{\mathcal{D}} \exp(\log(\alpha\gamma)) = \mathrm{OPT}_\gamma^{\mathcal{D}}\alpha\gamma.$$

473  So in either case, we have some sequence of $\mathbf{x}$s so that the projection onto $V^\perp$ of $\mathbf{Chow}(f_{\mathbf{w}^*}) - P$
474  is sufficiently small. This completes our analysis. $\qquad\square$

475  In order to extend this to a proof of Theorem A.3, we will need to reduce to solving the problem on a
476  finite sample set. This result can be obtained from Proposition A.4 by some fairly simple reductions.

477  Firstly, we note that we can assume that $\mathrm{OPT}_\gamma^{\mathcal{D}} \geq \epsilon/\alpha$, as increasing it to this value does not change
478  the problem.

479  Secondly, we note that if we let $\widehat{\mathcal{D}}$ be the empirical distribution over a set of $\Omega(d/\epsilon^2)$ random samples,
480  then with at least $2/3$ probability we have that

481    • $\mathbf{Pr}_{(\mathbf{x},y)\sim\widehat{\mathcal{D}}}[y\langle\mathbf{w}^*,\mathbf{x}\rangle \geq \gamma] \geq 1 - O(\mathrm{OPT}_\gamma^{\mathcal{D}})$

482    • For any vector $\mathbf{w}$, $\mathbf{Pr}_{(\mathbf{x},y)\sim\mathcal{D}}[\mathrm{sign}(\langle\mathbf{w},\mathbf{x}\rangle) \neq y] = \mathbf{Pr}_{(\mathbf{x},y)\sim\widehat{\mathcal{D}}}[\mathrm{sign}(\langle\mathbf{w},\mathbf{x}\rangle) \neq y] + O(\epsilon)$.

483  The first statement here is by applying the Markov inequality to the probability that $y\langle\mathbf{w}^*,\mathbf{x}\rangle < \gamma$,
484  and the second is by the VC-inequality [DL01]. We note that if the above hold, applying the algorithm
485  from Proposition A.4 to $\widehat{\mathcal{D}}$ will produce an appropriate $\mathbf{w}$. This produces an algorithm that uses
486  $O(d/\epsilon^2)$ samples and has runtime $O(d/\gamma\epsilon)^{\tilde{O}(1/(\alpha\gamma)^2)}$.

487  Unfortunately, this algorithm is not quite satisfactory as the runtime and sample complexity scale
488  poorly with the dimension $d$. In order to fix this, we will make use of an idea from [KS04]. Namely,
489  we will first apply dimension reduction to a smaller number of dimensions before applying our
490  algorithm. In particular, we will make use of the Johnson-Lindenstrauss lemma:

491  **Lemma A.8** ([JL84]). *There exists a probability distribution over linear transformations $A : \mathbb{R}^d \to$*
492  $\mathbb{R}^m$ *with $m = O(\log(1/\delta)/\epsilon^2)$ so that for any unit vectors $\mathbf{v},\mathbf{w} \in \mathbb{R}^d$, $\mathbf{Pr}_A[|\langle\mathbf{v},\mathbf{w}\rangle - \langle A\mathbf{v}, A\mathbf{w}\rangle| >$*
493  $\epsilon] < \delta$. *Additionally, there are efficient algorithms to sample from such distributions over $A$.*

494  We note that this implies in particular that $\|A\mathbf{v}\|_2 = 1 \pm \epsilon$ except for with probability $\delta$. Thus, by
495  tweaking the parameters a little bit and letting $h_A(\mathbf{v}) = A\mathbf{v}/\|A\mathbf{v}\|_2$, we have that $h_A(\mathbf{v})$ is always a
496  unit vector and that $\langle h_A(\mathbf{v}), h_A(\mathbf{w})\rangle = \langle\mathbf{v},\mathbf{w}\rangle \pm \epsilon$ except with probability $\delta$.

497  Next, we note that by taking $\epsilon = \gamma/2$ and $\delta = \mathrm{OPT}_\gamma^{\mathcal{D}}$ in the above we have that

$$\mathbf{Pr}_{A,(\mathbf{x},y)\sim\mathcal{D}}[y\langle h_A(\mathbf{w}^*), h_A(\mathbf{x})\rangle < \gamma/2]$$
$$\leq \mathbf{Pr}_{(\mathbf{x},y)\sim\mathcal{D}}[y\langle\mathbf{w}^*,\mathbf{x}\rangle < \gamma] + \mathbf{Pr}_{A,(\mathbf{x},y)\sim\mathcal{D}}[|\langle h_A(\mathbf{w}^*), h_A(\mathbf{x})\rangle - \langle\mathbf{w}^*,\mathbf{x}\rangle| > \gamma/2]$$
$$= O(\mathrm{OPT}_\gamma^{\mathcal{D}}).$$

Thus, by the Markov inequality, with large constant probability over $A$ we have that

$$\mathbf{Pr}_{(\mathbf{x},y)\sim\mathcal{D}}[y\langle h_A(\mathbf{w}^*), h_A(\mathbf{x})\rangle < \gamma/2] = O(\mathrm{OPT}_\gamma^{\mathcal{D}}).$$

But this means that the distribution $(h_A(\mathbf{x}), y)$ satisfies the assumptions for our algorithm (with $\gamma$
replaced by $\gamma/2$ and $\mathrm{OPT}_\gamma^{\mathcal{D}}$ by $O(\mathrm{OPT}_\gamma^{\mathcal{D}})$), but in dimension $m = O(\log(\alpha/\epsilon)/\gamma^2)$. Running the
algorithm described above on this set will find us a vector $\mathbf{w}$ so that

$$\mathbf{Pr}_{(\mathbf{x},y)\sim\mathcal{D}}[\mathrm{sign}(\langle\mathbf{w}, h_A(\mathbf{x})\rangle) \neq y] = O(\alpha\mathrm{OPT}_\gamma^{\mathcal{D}} + \epsilon).$$

However, it should be noted that

$$\mathrm{sign}(\langle\mathbf{w}, h_A(\mathbf{x})\rangle) = \mathrm{sign}(\langle\mathbf{w}, A\mathbf{x}\rangle/\|A\mathbf{x}|_2) = \mathrm{sign}(\langle\mathbf{w}, A\mathbf{x}\rangle) = \mathrm{sign}(A^T\langle\mathbf{w},\mathbf{x}\rangle).$$

498  Thus, $A^T\mathbf{w}$ satisfies the necessary conditions.

499  Our final algorithm is given below:

---

**Algorithm 4** $\alpha$-Agnostic Proper Learning of Theorem A.3

---

1: Pick $A : \mathbb{R}^d \to \mathbb{R}^m$ with $m = O(\log(\alpha/\epsilon)/\gamma^2)$ from an appropriate Johnson-Lindenstrauss family and define $f_A$ appropriately.
2: Take $O(m/\epsilon^2)$ random samples and let $\widehat{\mathcal{D}}$ be the uniform distribution over $(A\mathbf{x}/\|A\mathbf{x}|_2, y)$ for samples $(\mathbf{x}, y)$ from this set.
3: Run the algorithm from Proposition A.4 on $\widehat{\mathcal{D}}$ using $\gamma/2$ instead of $\gamma$ to find a vector $\mathbf{w}$.
4: **return** $A^T \mathbf{w}$.

---

## B  Omitted Results from Section 3

### B.1  Lower Bounds with Stronger Quantifiers on Parameters

Before we proceed to our proofs, let us first state a running time lower bound with stronger quantifier. Recall that previously we only rule out algorithms that work *for all* combinations of $d, \gamma, \varepsilon$. Below we relax the quantifier so that we need the *for all* quantifier only for $d$.

**Lemma B.1.** *Assuming the (randomized) ETH, for any universal constant $\alpha \geq 1$, there exists $\varepsilon_0 = \varepsilon_0(\alpha)$ such that there is no $\alpha$-agnostic learner for $\gamma$-margin halfspaces that runs in time $O(2^{(1/\gamma)^{2-o(1)}})\mathrm{poly}(d)$ for all $d$ and for some $0 < \varepsilon < \varepsilon_0$ and $\frac{1}{d^{0.5-o(1)}} \leq \gamma = \gamma(d) \leq \frac{1}{(\log d)^{0.5+o(1)}}$ that satisfies $\frac{\gamma(d+1)}{\gamma(d)} \geq \Omega(1)$.*

We remark here that the lower and upper bounds on $\gamma$ are essentially (i.e., up to lower order terms) the best possible. On the upper bound front, if $\gamma \geq \tilde{O}\left(\frac{1}{\sqrt{\log d}}\right)$, then our algorithmic result (Theorem 2.1) already give a $\mathrm{poly}(d, \frac{1}{\varepsilon})$-time $\alpha$-agnostic learner for $\gamma$-margin halfspaces (for all constant $\alpha > 1$). On the other hand, if $\gamma \leq O(\frac{1}{d^{0.5+o(1)}})$, then the trivial algorithm that exactly solves ERM for $m = O\left(\frac{d}{\varepsilon^2}\right)$ samples only takes $2^{O(d/\varepsilon^2)}$ time, which is already asymtotically faster than $2^{(1/\gamma)^{2-o(1)}}$. The last condition that $\frac{\gamma(d+1)}{\gamma(d)}$ is not too small is a sanity-check condition that prevents "sudden jumps" in $\gamma(d)$ such as $\gamma(d) = \frac{1}{(\log d)^{0.1}}$ and $\gamma(d+1) = \frac{1}{(d+1)^{0.1}}$; note that the condition is satisfied by "typical functions" such as $\gamma(d) = \frac{1}{d^c}$ or $\gamma(d) = \frac{1}{(\log d)^c}$ for some constant $c$.

As for $\varepsilon$, we only require the algorithm to work for any $\varepsilon$ that is not *too large*, i.e., no larger than $\varepsilon_0(\alpha)$. This later number is just a constant (when $\alpha$ is a constant). We note that it is still an interesting open question to make this requirement as mild as possible; specifically, is it possible to only require the algorithm to work for any $\varepsilon < 1/2$?

### B.2  Reduction from $k$-Clique and Proof of Theorem 3.2

We now proceed to the proofs of our results, starting with Theorem 3.2.

To prove Theorem 3.2, we reduce from the $k$-Clique problem. In $k$-Clique, we are given a graph $G$ and an integer $k$, and the goal is to determine whether the graph $G$ contains a $k$-clique (as a subgraph).

We take the perspective of *parameterized complexity*. Recall that a parameterized problem with parameter $k$ is said to be *fixed parameter tractable (FPT)* if it can be solved in time $f(k)\mathrm{poly}(n)$ for some computable function $f$, where $n$ denotes the input size.

It is well-known that $k$-Clique is complete for the class W[1] [DF95]. In other words, under the (widely-believed) assumption that W[1] does not collapse to FPT (the class of fixed parameter tractable problems), we cannot solve $k$-Clique in time $f(k)\mathrm{poly}(n)$ for any computable function $f$. We shall not formally define the class W[1] here; interested readers may refer to the book of Downey and Fellows for a much more in-depth discussion of the topic [DF13].

Our reduction starts with an instance of $k$-Clique and produces an instance of agnostic learning with margin $\gamma$ such that $\gamma = \Omega(1/k)$ (and the dimension is polynomial):

**Lemma B.2.** *There exists a polynomial-time algorithm that takes as input an $n$-vertex graph instance $G$ and an integer $k$, and produces a distribution $\mathcal{D}$ over $\mathbb{B}_d \times \{\pm 1\}$ and $\gamma, \kappa \in [0, 1]$ such that*

537　　　　　　• *(Completeness)* If $G$ contains a $k$-clique, then $\mathrm{OPT}^{\mathcal{D}}_{\gamma} \leq \kappa$.

538　　　　　　• *(Soundness)* If $G$ does not contains a $k$-clique, then $\mathrm{OPT}^{\mathcal{D}}_{0-1} > \kappa + \frac{0.001}{n^3}$.

539　　　　　　• *(Margin Parameter)* $\gamma \geq \Omega(\frac{1}{\sqrt{k}})$.

540　We remark here that, in Lemma B.2 and throughout the remainder of the section, we say that an
541　algorithm produces a distribution $\mathcal{D}$ over $\mathbb{B}_d \times \{\pm 1\}$ to mean that it outputs the set of samples
542　$\{(\mathbf{x}^{(i)}, y^{(i)})\}_{i \in [m]}$ and numbers $d_i$ for each $i \in [m]$ representing the probability of $(\mathbf{x}^{(i)}, y^{(i)})$ with
543　respect to $\mathcal{D}$. Note that this is stronger than needed since, to prove hardness of learning, it suffices to
544　have an oracle that can sample from $\mathcal{D}$, but here we actually explicitly produce a full description of
545　$\mathcal{D}$. Moreover, note that this implicitly implies that the support of $\mathcal{D}$ is of polynomial size (and hence,
546　for any given $h$, $\mathrm{err}^{\mathcal{D}}_{\gamma}(h)$ and $\mathrm{err}^{\mathcal{D}}_{0-1}(h)$ can be efficiently computed).

547　As stated above, Lemma B.2 immediately implies Theorem 3.2 because, if we can agnostically learn
548　$\gamma$-margin halfspaces in time $f(\frac{1}{\gamma})\mathrm{poly}(d, \frac{1}{\varepsilon})$, then we can solve $k$-Clique in $f(O(\sqrt{k}))\mathrm{poly}(n)$ time,
549　which would imply that W[1] is contained in (randomized) FPT. This is formalized below.

550　*Proof of Theorem 3.2.* Suppose that we have an $f(\frac{1}{\gamma})\mathrm{poly}(d, \frac{1}{\varepsilon})$-time agnostic learner for $\gamma$-margin
551　halfspaces. Given an instance $(G, k)$ of $k$-Clique, we run the reduction from Lemma B.2 to produce
552　a distribution $\mathcal{D}$. We then run the learner on $\mathcal{D}$ with $\varepsilon = \frac{0.001}{n^3}$ (and with $\delta = 1/3$). Note that the
553　learner runs in time $f(O(\sqrt{k}))\mathrm{poly}(n)$ and produces a halfspace $h$. We then compute $\mathrm{err}^{\mathcal{D}}_{0-1}(h)$; if
554　it is no more than $\kappa + \frac{0.001}{n^3}$, then we output YES. Otherwise, output NO.

555　The algorithm described above solves $k$-Clique (correctly with probability 2/3) in FPT time. Since
556　$k$-Clique is W[1]-complete, this implies that W[1] is contained in randomized FPT.　　　□

557　We now move on to prove Lemma B.2. Before we do so, let us briefly describe the ideas behind it.
558　The dimension $d$ will be set to $n$, the number of vertices of $G$. Each coordinate $\mathbf{w}_i$ is associated with
559　a vertex $i \in V(G)$. In the completeness case, we would like to set $\mathbf{w}_i = \frac{1}{\sqrt{k}}$ iff $i$ is in the $k$-clique
560　and $\mathbf{w}_i = 0$ otherwise. To enforce a solution to be of this form, we add two types of samples that
561　induces the following constraints:

562　　　　　　• *Non-Edge Constraint:* for every *non*-edge $(i, j)$, we should have $\mathbf{w}_i + \mathbf{w}_j \leq \frac{1}{\sqrt{k}}$. That is,
563　　　　　　we should "select" at most one vertex among $i, j$.

564　　　　　　• *Vertex Selection Constraint:* each coordinate of $\mathbf{w}$ is at least $\frac{1}{\sqrt{k}}$. Note that we will violate
565　　　　　　such constraints for all vertices, except those that are "selected".

566　If we select the probabilities in $\mathcal{D}$ so that the non-edge constraints are weighted much larger than the
567　vertex selection constraints, then it is always better to not violate any of the first type of constraints.
568　When this is the case, the goal will now be to violate as few vertex selection constraints as possible,
569　which is the same as finding a maximum clique, as desired.

570　While the above paragraph describes the core idea of the reduction, there are two additional issues
571　we have to resolve:

572　　　　　　• *Constant Coordinate:* first, notice that we cannot actually quite write a constraint of the
573　　　　　　form $\mathbf{w}_i + \mathbf{w}_j \leq \frac{1}{\sqrt{k}}$ using the samples because there is no way to express a value like $\frac{1}{\sqrt{k}}$
574　　　　　　directly. To overcome this, we have a "constant coordinate" $\mathbf{w}_*$, which is supposed to be
575　　　　　　a constant, and replace the right hand side of non-edge constraints by $\frac{\mathbf{w}_*}{\sqrt{k}}$ (instead of $\frac{1}{\sqrt{k}}$).
576　　　　　　The new constraint can now be represented by a sample.

577　　　　　　• *Margin:* in the above reduction, there was no margin at all! To get the appropriate margin,
578　　　　　　we "shift" the constraint slightly so that there is a margin. For instance, instead of $\frac{\mathbf{w}_*}{\sqrt{k}}$ for a
579　　　　　　non-edge constraint, we use $\frac{1.1\mathbf{w}_*}{\sqrt{k}}$. We now have a margin of $\approx \frac{0.1}{\sqrt{k}}$ and it is still possible
580　　　　　　to argue that the best solution is still to select a clique.

581　The reduction, which follows the above outline, is formalized below.

*Proof of Lemma B.2.* Given a graph $G = (V, E)$, we use $n$ to denote the number of vertices $|V|$ and we rename its vertices so that $V = [n]$. We set $d = n + 1$; we name the first coordinate $*$ and each of the remaining coordinates $i \in [n]$. For brevity, let us also define $\beta = 1 - \frac{0.01}{n^2}$. The distribution $\mathcal{D}$ is defined as follows:

- Add a labeled sample $(-\mathbf{e}^*, -1)$ with probability $\frac{\beta}{2}$ in $\mathcal{D}$. We refer to this as the *positivity constraint for* $*$.

- For every pair of distinct vertices $i, j$ that do not induce an edge in $E$, add a labeled sample $(\frac{1}{2}\left(\frac{1.1}{\sqrt{k}}\mathbf{e}^* - \mathbf{e}^i - \mathbf{e}^j\right), 1)$ with probability $\frac{\beta}{2(\binom{n}{2} - |E|)}$ in $\mathcal{D}$. We refer to this as the *non-edge constraint for* $(i, j)$.

- For every vertex $i$, add a labeled sample $(\frac{1}{2}\left(\mathbf{e}^i - \frac{0.9}{\sqrt{k}}\mathbf{e}^*\right), 1)$ with probability $\frac{0.01}{n^3}$ in $\mathcal{D}$. We refer to this as the *vertex selection constraint for* $i$.

Finally, let $\gamma = \frac{0.1}{2\sqrt{2k}}$, $\kappa = (n - k)\left(\frac{0.01}{n^3}\right)$. It is obvious that the reduction runs in polynomial time.

**Completeness.** Suppose that $G$ contains a $k$-clique; let $S \subseteq V$ denote the set of its vertices. We define $\mathbf{w}$ by $\mathbf{w}_* = \frac{1}{\sqrt{2}}$ and, for every $i \in V$, $\mathbf{w}_i = \frac{1}{\sqrt{2k}}$ if $i \in S$ and $\mathbf{w}_i = 0$ otherwise. It is clear that $\|\mathbf{w}\|_2 = 1$ and that, for every $(\mathbf{x}, y) \in \mathsf{supp}(\mathcal{D})$, we have $|\langle \mathbf{w}, \mathbf{x}\rangle| \geq \frac{0.1}{2\sqrt{2k}}$. Finally, observe that all the first two types of constraints are satisfied, and a vertex selection constraint for $i$ is unsatisfied iff $i \notin S$. Thus, we have $\mathrm{err}_\gamma^{\mathcal{D}}(\mathbf{w}) = (n - k)\left(\frac{0.01}{n^3}\right) = \kappa$, which implies that $\mathrm{OPT}_\gamma^{\mathcal{D}} \leq \kappa$ as desired.

**Soundness.** Suppose contrapositively that $\mathrm{OPT}_{0-1}^{\mathcal{D}} \leq \kappa + \frac{0.001}{n^3}$; that is, there exists $\mathbf{w}$ such that $\mathrm{err}_{0-1}^{\mathcal{D}}(\mathbf{w}) \leq \kappa + \frac{0.001}{n^3}$. Observe that each labeled sample of the first two types of constraints has probability more than $\frac{\beta}{2n^2} > \kappa + \frac{0.001}{n^3}$. As a result, $\mathbf{w}$ must correctly classifies these samples. Since $\mathbf{w}$ correctly classifies $(-\mathbf{e}^*, -1)$, it must be that $w_* > 0$.

Now, let $T$ be the set of vertices $i$ such that $\mathbf{w}$ mislabels the vertex selection constraint for $i$. Observe that $|T| < \frac{\left(\kappa + \frac{0.001}{n^3}\right)}{\frac{0.01}{n^3}} < n - k + 1$. In other words, $S := V \setminus T$ is of size at least $k$. We claim that $S$ induces a $k$-clique in $G$. To see that this is true, consider a pair of distinct vertices $i, j \in S$. Since $\mathbf{w}$ satisfies the vertex selection constraints for $i$ and for $j$, we must have $\mathbf{w}_i, \mathbf{w}_j \geq \frac{0.9}{\sqrt{k}}$. This implies that $(i, j)$ is an edge, as otherwise $\mathbf{w}$ would mislabel the non-edge constraint for $(i, j)$.

As a result, $G$ contains a $k$-clique as desired. $\square$

## B.3 Reduction from $k$-CSP and Proofs of Theorems 3.1, 3.3 and Lemma B.1

In this section, we will prove Theorems 3.1 and 3.3, by reducing from the hardness of approximation of constraint satisfaction problems (CSPs), given by PCP Theorems.

### B.3.1 CSPs and PCP Theorem(s)

Before we can state our reductions, we have to formally define CSPs and state the PCP theorems we will use more formally. We start with the definition of $k$-CSP:

**Definition B.3** ($k$-CSP). For any integer $k \in \mathbb{N}$, a $k$-CSP instance $\mathcal{L} = (V, \Sigma, \{\Pi_q\}_{q \in \mathcal{Q}})$ consists of

- The variable set $V$,

- The alphabet $\Sigma$, which we sometimes refer to as labels,

- Constraints set $\{\Pi_S\}_{S \in \mathcal{Q}}$, where $\mathcal{Q} \subseteq \binom{V}{k}$ is a collection of $k$-size subset of $V$. For each subset $S = \{v_1, \ldots, v_k\}$, $\Pi_S \subseteq \Sigma^S$ is the set of accepting answers for the constraint $\Pi_S$. Here we think of each $f \in \Sigma^S$ as a function from $f : S \to \Sigma$.

A $k$-CSP instance is said to be *regular* if each variable appears in the same number of constraints.

622 An assignment $\phi$ is a function $\phi : V \to \Sigma$. Its value, denoted by $\mathrm{val}_{\mathcal{L}}(\phi)$, is the fraction of constraints
623 $S \in \mathcal{Q}$ such that[4] $\phi|_S \in \Pi_S$. Such constraints are said to be *satisfied by* $\phi$. The value of $\mathcal{L}$, denoted
624 by $\mathrm{val}(\mathcal{L})$, is the maximum value among all assignments, i.e., $\mathrm{val}(\mathcal{L}) := \max_{\phi} \mathrm{val}_{\mathcal{L}}(\phi)$.

625 In the $\nu$-GAP-$k$-CSP problem, we are given a regular instance $\mathcal{L}$ of $k$-CSP, and we want to distinguish
626 between $\mathrm{val}(\mathcal{L}) = 1$ and $\mathrm{val}(\mathcal{L}) < \nu$.

627 Throughout this subsection, we use $n$ to denote the instance size of $k$-CSP, that is $n = \sum_{S \in \mathcal{Q}} |\Pi_S|$.

628 The celebrated PCP theorem [AS98, ALM+98] is equivalent to the proof of NP-hardness of approxi-
629 mating $\nu$-Gap-$k$-CSP for some constant $k$ and $\nu < 1$. Since we would like to prove (tight) running
630 time lower bounds, we need the versions of PCP Theorems that provides strong running time lower
631 bounds as well. For this task, we turn to the Moshkovitz-Raz PCP theorem, which can not only
632 achieve arbitrarily small constant $\nu > 0$ but also almost exponential running time lower bound.

633 **Theorem B.4** (Moshkovitz-Raz PCP [MR10]). *Assuming ETH, for any $0 < \nu < 1$, $\nu$-Gap-2-CSP*
634 *cannot be solved in time $O(2^{n^{1-o(1)}})$, even for instances with $|\Sigma| = O_\nu(1)$.*

635 As for our hardness of approximation result (Theorem 3.3), we are aiming to get as large a ratio as
636 possible. For this purpose, we will use a PCP Theorem of Dinur, Harsha and Kindler, which achieves
637 $\nu = \frac{1}{\mathrm{poly}(n)}$ but need $k$ to be polyloglog($n$).

638 **Theorem B.5** (Dinur-Harsha-Kindler PCP [DHK15]). *$n^{-\Omega(1)}$-Gap-polyloglog($n$)-CSP is NP-hard.*

639 Finally, we state the Sliding Scale Conjecture (SSC) of Bellare et al. [BGLR94], which says that the
640 NP-hardness with $\nu = \frac{1}{\mathrm{poly}(n)}$ holds even in the case where $k$ is constant:

641 **Conjecture 1** (Sliding Scale Conjecture [BGLR94]). *For some constant $k$, $n^{-\Omega(1)}$-Gap-$k$-CSP is*
642 *NP-hard.*

643 ### B.3.2 Reducing from $k$-CSP to Agnostically Learning Halfspaces with Margin

644 Having set up the notation, we now move on to the reduction from $k$-CSP to agnostic learning of half-
645 spaces with margin. Our reduction can be viewed as a modification of the reduction from [ABSS97];
646 compared to [ABSS97], we have to (1) be more careful so that we can get the margin in the com-
647 pleteness and (2) modify the reduction to work even for $k > 2$.

648 Before we precisely state the formal properties of the reduction, let us give a brief informal intuition
649 behind the reduction. Given an instance $\mathcal{L} = (V, \Sigma, \{\Pi_S\}_{S \in \mathcal{Q}})$ of $k$-CSP, we will create a distribution
650 $\mathcal{D}$ on $\mathbb{B}_d \times \{\pm 1\}$, where the dimension $d$ is equal to $n$. Each coordinate is associated with an
651 accepting answer of each constraint; that is, each coordinate is $(S, f)$ where $S \in \mathcal{Q}$ and $f \in \Pi_S$. In
652 the completeness case where we have a perfect assignment $\phi$, we would like the halfspace's normal
653 vector to set $\mathbf{w}_{(S,f)} = 1$ iff $f$ is the assignment to predicate $S$ in $\phi$ (i.e., $f = \phi|_S$), and zero otherwise.
654 To enforce this, we add three types of constraints:

- *Non-negativity Constraint*: that each coordinate of $\mathbf{w}$ should be non-negative.

- *Satisfiability Constraint*: that for each $S \in \mathcal{Q}$, $\mathbf{w}_{(S,f)}$ is positive for at least one $f \in \Pi_S$.

- *Selection Constraint*: for each variable $v \in V$ and label $\sigma \in \Sigma$, we add a constraint that the
  sum of all $\mathbf{w}_{(S,f)}$, for all $S$ that $v$ appears in and all $f$ that assigns $\sigma$ to $v$, is non-positive.

659 Notice that, for the completeness case, we satisfy the first two types of constraints, and we violate the
660 selection constraints only when $\phi(v) = \sigma$. Intuitively, in the soundness case, we will not be able to
661 "align" the positive $\mathbf{w}_{(S,f)}$ from different $S$'s together, and we will have to violate a lot more selection
662 constraints.

663 Of course, there are many subtle points that the above sketch does not address, such as the margin; on
664 this front, we add one more special coordinate $\mathbf{w}_*$, which we think of as being equal to 1, and we
665 add/subtract $\delta$ times this coordinate to each of the constraints, which will create the margin for us.
666 Another issue is that the normal vector of the halfspace (and samples) as above have norm more than
667 one. Indeed, our assignment in the completeness case has norm $O(\sqrt{n})$. Hence, we have to scale the

668  normal vector down by a factor of $O(\sqrt{n})$, which results in a margin of $\gamma = \Omega(1/\sqrt{n})$. This is the
669  reason why we arrive at the running time lower bound of the form $2^{\gamma^{2-o(1)}}$.

670  The properties and parameter dependencies of the reduction are encapsulated in the following theorem.

671  **Theorem B.6.** *There exists a polynomial time reduction that takes as input a regular instance*
672  $\mathcal{L} = (V, \Sigma, \{\Pi_S\}_{S \in \mathcal{Q}})$ *of $k$-CSP and a real number $\nu > 0$, and produces a distribution $\mathcal{D}$ over*
673  $\mathbb{B}_d \times \{\pm 1\}$ *and positive real numbers $\gamma, \kappa, \varepsilon, \alpha$ such that*

674  - *(Completeness) If $\mathcal{L}$ is satisfiable, then $\mathrm{OPT}_\gamma^{\mathcal{D}} \leq \kappa$.*

675  - *(Soundness) If $val(\mathcal{L}) < \nu$, then $\mathrm{OPT}_{0-1}^{\mathcal{D}} > \alpha \cdot \kappa + \varepsilon$.*

676  - *(Margin Parameter) $\gamma = \Omega\left(\frac{1}{\Delta|\Sigma|^{3k}\sqrt{|\mathcal{Q}|}}\right)$, where $\Delta$ denotes the number of constraints*
677  *each variable appears in.*

678  - *(Approximation Ratio) $\alpha = \Omega\left(\frac{(1/\nu)^{1/k}}{k}\right)$.*

679  - *(Error Parameter) $\varepsilon = \Omega\left(\frac{1}{\Delta|\Sigma|^k}\right) \cdot \alpha$.*

680  - *(Dimension) $d = n + 1$.*

681  *Proof.* Before we define $\mathcal{D}$, let us specify the parameters:

682  - First, we let $d$ be $1 + n$. We name the first coordinate as $*$ and each of the remaining
683  coordinates are named $(S, f)$ for a constraint $S \in \mathcal{Q}$ and $f \in \Pi_S$.

684  - Let $Z := 2\left(|V| \cdot |\Sigma| + 2k|\mathcal{Q}| + 2k\sum_{e \in E}|\Pi_e|\right)$ be our "normalizing factor", which will
685  be used below to normalized the probability.

686  - Let $\delta := \frac{0.1}{\Delta|\Sigma|^{2k}}$ be the "shift parameter". Note that this is not the margin $\gamma$ (which will be
687  defined below).

688  - Let $s := 10\Delta|\Sigma|^k$ be the scaling factor, which we use to make sure that all our samples lie
689  within the unit ball.

690  - Let the gap parameter $\alpha$ be $\frac{(1/\nu)^{1/k}}{40k}$.

691  - Finally, let $\kappa = \frac{|V|}{Z}$ and $\varepsilon = \kappa \cdot \alpha$.

692  Note that $\alpha$ as defined above can be less than one. However, this is not a problem: in the subsequent
693  proofs of Theorems 3.1 and 3.3, we will always choose the settings of parameters so that $\alpha > 1$.

694  We are now ready to define the distribution $\mathcal{D}$ on $\mathbb{B}_d \times \{\pm 1\}$, as follows:

695  1. Add a labeled sample $(-\mathbf{e}^*, -1)$ with probability $1/2$ to $\mathcal{D}$. This corresponds to the
696  constraint $\mathbf{w}_* > 0$; we refer to this as the *positivity constraint for $*$*.

697  2. Next, for each coordinate $(S, f)$, add a labeled sample $\left(\frac{1}{s}\left(\mathbf{e}^{(S,f)} + \delta \cdot \mathbf{e}^*\right), 1\right)$ with prob-
698  ability $2k/Z$ to $\mathcal{D}$. This corresponds to $\mathbf{w}_{(S,f)} + \delta \cdot \mathbf{w}_* \geq 0$ scaled down by a factor of
699  $1/s$ so that the vector is in the unit ball; we refer to this as the *non-negativity constraint for*
700  $(S, f)$.

701  3. For every $S \in \mathcal{Q}$, add a labeled sample $\left(\frac{1}{s}\left(\sum_{f \in \Pi_S}\mathbf{e}^{(S,f)} - (1-\delta)\mathbf{e}^*\right), 1\right)$ with proba-
702  bility $2k/Z$ to $\mathcal{D}$. This corresponds to the constraint $\sum_{f \in \Pi_S}\mathbf{w}_{(S,f)} \geq (1-\delta)\mathbf{w}_*$, scaled
703  down by a factor of $1/s$. We refer to this constraint as the *satisfiability constraint for $S$*.

4. For every variable $v \in V$ and $\sigma \in \Sigma$, add a labeled sample
$\left( \frac{1}{s} \left( \sum_{S \in \mathcal{Q}: v \in S} \sum_{f \in \Pi_S : f(v) = \sigma} \mathbf{e}^{(S,f)} - \delta \mathbf{e}^* \right), -1 \right)$ with probability $1/Z$ to $\mathcal{D}$. This corresponds to the constraint $\sum_{S \in \mathcal{Q}: v \in S} \sum_{f \in \Pi_S : f(v) = \sigma} \mathbf{w}_{(S,f)} < \delta \cdot \mathbf{w}_*$, scaled down by a factor of $1/s$. We refer to this as the *selection constraint for* $(v, \sigma)$.

**Completeness.** Suppose that there exists an assignment $\phi : V \to \Sigma$ that satisfies all the constraints of $\mathcal{L}$. Consider the halfspace with normal vector $\mathbf{w}$ defined by $\mathbf{w}_* = \zeta$ and

$$\mathbf{w}_{(S,f)} = \begin{cases} \zeta & \text{if } f = \phi|_S, \\ 0 & \text{otherwise,} \end{cases}$$

where $\zeta := \frac{1}{\sqrt{1+|\mathcal{Q}|}}$ is the normalization factor. It is easy to see that the positivity constraints and the satisfiability constraints are satisfied with margin at least $\gamma = \zeta \cdot \delta / s = \Omega \left( \frac{1}{\Delta |\Sigma|^{3k} \sqrt{|\mathcal{Q}|}} \right)$. Finally, observe that the sum $\sum_{S \in \mathcal{Q}: v \in S} \sum_{f \in \Pi_S : f(v) = \sigma} \mathbf{w}_{(S,f)}$ is zero if $f(v) \neq \sigma$; in this case, the selection constraint for $(v, \sigma)$ is also satisfied with margin at least $\gamma$. As a result, we only incur an error (with respect to margin $\gamma$) for the selection constraint for $(v, \phi(v))$ for all $v \in V$; hence, we have $\mathrm{err}_\gamma^{\mathcal{D}}(\mathbf{w}) \leq \frac{1}{Z} \cdot |V| = \kappa$ as desired.

**Soundness.** Suppose contrapositively that there exists $\mathbf{w}$ with $\mathrm{err}_{0-1}^{\mathcal{D}}(\mathbf{w}) \leq \alpha \cdot \kappa + \varepsilon = 2\alpha\kappa$. We will "decode" back an assignment with value at least $\nu$ of the CSP from $\mathbf{w}$.

To do so, first observe that from the positivity constraint for $*$, we must have $\mathbf{w}_* > 0$, as otherwise we would already incur an error of $1/2 > 2\alpha\kappa$ with respect to $\mathcal{D}$. Now, since scaling (by a positive factor) does not change the fraction of samples violated, we may assume w.l.o.g. that $\mathbf{w}_* = 1$.

Next, we further claim that we may assume without loss of generality that $\mathbf{w}$ does not violate any non-negativity constraints (2) or satisfiability constraints (3). The reason is that, if $\mathbf{w}$ violates a non-negativity constraint for $(S = \{v_1, \ldots, v_k\}, f)$, then we may simply change $\mathbf{w}_{(S,f)}$ to zero. This reduces the error by $2k/Z$, while it may only additionally violate $k$ additional selection constraints for $(v_1, f(v_1)), \ldots, (v_k, f(v_k))$ which weights $k/Z$ in total with respect to $\mathcal{D}$. As a result, this change only reduces the error in total. Similarly, if the satisfiability constraint of $S$ is unsatisfied, we may change $\mathbf{w}_{(S,f)}$ for some $f \in \Pi_S$ to a sufficiently large number so that this constraint is satisfied; once again, in total the error decreases. Hence, we may assume that the non-negativity constraints (2) and satisfiability constraints (3) all hold.

Now, for every vertex $v$, let $L_v \subseteq \Sigma$ denote the set of labels $\sigma \in \Sigma$ such that the selection constraint for $(v, \sigma)$ is violated. Since we assume that $\mathrm{err}_{0-1}^{\mathcal{D}}(\mathbf{w}) \leq 2\alpha\kappa$, we must have $\sum_{v \in V} |L_v| \leq (2\alpha\kappa)/(1/Z) = 2\alpha|V|$.

Next, let $V_{\text{small}}$ denote the set of all variables $v \in V$ such that $|L_v| \leq 20\alpha k$. From the bound we just derived, we must have $|V_{\text{small}}| \geq \left( 1 - \frac{1}{10k} \right) |V|$.

Another ingredient we need is the following claim:

**Claim B.7.** *For every constraint $S = \{v_1, \ldots, v_k\} \in \mathcal{Q}$, there exist $\sigma_1 \in L_{v_1}, \ldots, \sigma_k \in L_{v_k}$ that induces an accepting assignment for $\Pi_S$ (i.e., $f \in \Pi_S$ where $f$ is defined by $f(v_i) = \sigma_i$).*

*Proof.* Suppose for the sake of contradiction that no such $\sigma_1 \in L_{v_1}, \ldots, \sigma_k \in L_{v_k}$ exists. In other words, for every $f \in \Pi_S$, there must exist $i \in [k]$ such that the selection constraint for $(v_i, f(v_i))$ is not violated. This means that

$$\delta = \delta \cdot \mathbf{w}_* > \sum_{S' \in \mathcal{Q}: v \in S'} \sum_{f' \in \Pi_{S'} : f'(v) = \sigma} \mathbf{w}_{(S',f')}$$

$$\geq \mathbf{w}_{(S,f)} + \sum_{S' \in \mathcal{Q}: v \in S'} \sum_{f' \in \Pi_{S'} : f'(v) = \sigma} -\delta \cdot \mathbf{w}_*$$

$$\geq \mathbf{w}_{(S,f)} - \delta \cdot \Delta |\Sigma|^k$$

where the second inequality comes from our assumption, that the non-negativity constraints are satisfied.

743 Hence, by summing this up over all $f \in \Pi_S$, we get

$$\sum_{f \in \Pi_S} \mathbf{w}_{(S,f)} \leq \delta \cdot (\Delta |\Sigma|^k + 1) \cdot |\Sigma|^k < (1 - \delta),$$

744 which means that the satisfiability constraint for $S$ is violated, a contradiction. $\qquad \square$

745 We can now define an assignment $\phi : V \to \Sigma$ for $\mathcal{L}$ as follows. For every $v \in V$, let $\phi(v)$ be a random
746 label in $L_v$. Notice here that, by Claim B.7, the probability that a constraint $S = \{v_1, \ldots, v_k\}$ is
747 satisfied is at least $\prod_{i \in [k]} |L_{v_i}|^{-1}$. Hence, the expected total number of satisfied constraints is at least

$$\sum_{S=\{v_1,\ldots,v_k\} \in \mathcal{Q}} \prod_{i \in [k]} |L_{v_i}|^{-1} \geq \sum_{S=\{v_1,\ldots,v_k\} \in \mathcal{Q}: v_1,\ldots,v_k \in V_{\text{small}}} \prod_{i \in [k]} |L_{v_i}|^{-1}$$

$$\geq \sum_{S=\{v_1,\ldots,v_k\} \in \mathcal{Q}: v_1,\ldots,v_k \in V_{\text{small}}} (20\alpha k)^{-k}.$$

748 Recall that we have earlier bound $|V_{\text{small}}|$ to be at least $\left(1 - \frac{1}{10k}\right)|V|$. Hence, the fraction of
749 constraints that involves some variable outside of $V_{\text{small}}$ is at most $\left(\frac{1}{10k}\right) \cdot (k) = 0.1$. Plugging this
750 into the above inequality, we get that the expected total number of satisfied constraints is at least

$$0.9 |\mathcal{Q}| \cdot (20\alpha k)^{-k} > |\mathcal{Q}| \cdot \nu,$$

751 where the equality comes from our choice of $\alpha$. In other words, we have $\text{val}(\mathcal{L}) > \nu$ as desired. $\quad \square$

752 ### B.3.3 Proofs of Theorems 3.1, 3.3 and Lemma B.1

753 We now prove Theorem 3.1, by simply applying Theorem B.6 with appropriate parameters on top of
754 the Moshkovitz-Raz PCP.

755 *Proof of Theorem 3.1.* Suppose contrapositively that, for some constant $\tilde{\alpha} \geq 1$ and $\zeta > 0$, we have
756 an $O(2^{(1/\gamma)^{2-\zeta}} 2^{d^{1-\zeta}}) f(\frac{1}{\varepsilon})$ time $\tilde{\alpha}$-agnostic proper learner for $\gamma$-margin halfspaces.

757 Let $\nu > 0$ be a sufficiently small constant so that the parameter $\alpha$ (when $k = 2$) from Theorem B.6 is
758 at least $\tilde{\alpha}$. (In particular, we pick $\nu = \frac{1}{C(\tilde{\alpha})^k}$ for some sufficiently large constant $C$.)

759 Given an instance $\mathcal{L}$ of $\nu$-Gap-2-CSP, we run the reduction from Theorem B.6 to produce a distribution
760 $\mathcal{D}$. We then run the learner on $\mathcal{D}$ with error parameter $\varepsilon$ as given by Theorem B.6 (and with $\delta = 1/3$).
761 Note that the learner runs in $O(2^{(1/\gamma)^{2-\zeta}} 2^{d^{1-\zeta}}) f(\frac{1}{\varepsilon}) = 2^{O(n^{1-\zeta/2})}$ time, and produces a halfspace $h$.
762 We compute $\text{err}_{0-1}^{\mathcal{D}}(h)$; if it is no more than $\alpha \cdot \kappa + \varepsilon$, then we output YES. Otherwise, output NO.

763 The algorithm describe above solves $\nu$-Gap-2-CSP (correctly with probability 2/3) in $2^{O(n^{1-\zeta/2})}$
764 time, which, by Theorem B.4, violates (randomized) ETH. $\qquad \square$

765 Next, we prove Lemma B.1. The main difference from the above proof is that, since the algorithm
766 works only *for some* margin $\gamma = \gamma(d)$. We will select the dimension $d$ to be as large as possible so
767 that $\gamma(d)$ is still smaller than the margin given by Theorem B.6. This dimension $d$ will be larger than
768 the dimension given by Theorem B.6; however, this is not an issue since we can simply "pad" the
769 remaining dimensions by setting the additional coordinates to zeros. This is formalized below.

770 *Proof of Lemma B.1.* Let $\tilde{\alpha} \geq 1$ be any constant. Let $\nu > 0$ be a sufficiently small constant so that
771 the parameter $\alpha$ (when $k = 2$) from Theorem B.6 is at least $\tilde{\alpha}$. (In particular, we pick $\nu = \frac{1}{C(\tilde{\alpha})^k}$ for
772 some sufficiently large constant $C$.) Let $\varepsilon_0 = \varepsilon_0(\tilde{\alpha})$ be the parameter $\varepsilon$ given by Theorem B.6.

773 Suppose contrapositively that, for some $\zeta > 0$, there is an $\tilde{\alpha}$-agnostic learner $\mathcal{A}$ for $\gamma(\tilde{d})$-margin
774 halfspaces that runs in time $O(2^{(1/\gamma)^{2-\zeta}}) \text{poly}(\tilde{d})$ for all dimensions $\tilde{d}$ and for some $0 < \varepsilon^* < \varepsilon_0(\alpha)$
775 and $\gamma(\tilde{d})$ that satisfies

$$\frac{1}{\tilde{d}^{0.5-\zeta}} \leq \gamma(\tilde{d}) \leq \frac{1}{(\log \tilde{d})^{0.5+\zeta}} \qquad (4)$$

776    and

$$\frac{\gamma(\tilde{d}+1)}{\gamma(\tilde{d})} \geq \zeta. \tag{5}$$

777    We may assume without loss of generality that $\zeta < 0.1$.

778    We create an algorithm $\mathcal{B}$ for $\nu$-Gap-2-CSP as follows:

779    • Given an instance $\mathcal{L}$ of $\nu$-Gap-2-CSP of size $n$, we first run the reduction from Theorem B.6
780    with $\nu$ as selected above to produce a distribution $\mathcal{D}$ on $\mathbb{B}_d \times \{\pm 1\}$ (where $d = n + 1$). Let
781    the margin parameter $\gamma$ be as given in Theorem B.6; observe that $\gamma = \Omega_\nu(1/\sqrt{n})$.

782    • Let $\tilde{d}$ be the largest intger so that $\gamma(\tilde{d}) \geq \gamma$. Observe that, from the lower bound in (5),
783    we have $\gamma(d) \geq \frac{1}{d^{0.5-\zeta}}$. Hence, for a sufficiently large $d$, $\gamma(d)$ is larger than $\gamma$ (which is
784    $O_\nu(1/\sqrt{d})$). In other words, we have $\tilde{d} \geq d$.

785    • Create a distribution $\mathcal{D}'$ as follows: for each $(\mathbf{x}, y) \in \mathsf{supp}(\mathcal{D})$, we create a sample $(\mathbf{x}', y)$
786    in $\mathcal{D}'$ with the same probability and where $\mathbf{x}' \in \mathbb{B}_{\tilde{d}}$ is $\mathbf{x}$ concatenated with 0s in the last
787    $\tilde{d} - d$ coordinates.

788    • Run the learner $\mathcal{A}$ on $\mathcal{D}'$ with parameter $\gamma(\tilde{d})$ and $\varepsilon$. Suppose that it outputs a halfspace $h$.
789    We compute $\mathrm{err}_{0-1}^{\mathcal{D}'}(h)$; if this is no more than $\alpha \cdot \kappa + \varepsilon_0(\alpha)$, then output YES. Otherwise,
790    output NO.

791    It is simple to see that, in the completeness case, we must have $\mathrm{OPT}_{\gamma(\tilde{d})}^{\mathcal{D}'} \leq \mathrm{OPT}_\gamma^{\mathcal{D}'} = \mathrm{OPT}_\gamma^{\mathcal{D}} \leq \kappa$;
792    hence, $\mathcal{A}$ would (with probability 2/3) output a halfspace $h$ with 0-1 error at most $\alpha \cdot \kappa + \varepsilon_0(\alpha)$, and we
793    output YES. On the other hand, in the soundness case, we have $\mathrm{OPT}_{0-1}^{\mathcal{D}'} = \mathrm{OPT}_{0-1}^{\mathcal{D}'} > \alpha \cdot \kappa + \varepsilon_0(\tilde{\alpha})$,
794    and we always output NO. Hence, the algorithm is correct with probability 2/3.

795    Next, to analyze the running time of $\mathcal{B}$, let us make a couple additional observations. First, from (5),
796    we have

$$\gamma(\tilde{d}) \leq \gamma/\zeta \leq O(1/\sqrt{n}). \tag{6}$$

797    Furthermore, from the upper bound in (5), we have

$$\tilde{d} \leq 2^{(1/\gamma(\tilde{d}))^{\frac{1}{0.5+\zeta}}} \leq 2^{O(n^{\frac{1}{1+2\zeta}})} \leq 2^{O(n^{1-\zeta})}, \tag{7}$$

798    where the last inequality follows from $\zeta < 0.1$.

799    As a result, the algorithm runs in time $O(2^{(1/\gamma(d))^{2-\zeta}})\mathrm{poly}(\tilde{d}) \leq 2^{O(n^{1-\zeta/2})}$, which from Theo-
800    rem B.4 would break the (randomized) ETH.    □

801    Finally, we prove Theorem 3.3, which is again by simply applying Theorem B.6 to the Dinur-Harsha-
802    Kindler PCP and the Sliding Scale Conjecture.

803    *Proof of Theorem 3.3.* By plugging in our reduction from Theorem 3.3 to Theorem B.5, we get
804    that it is NP-hard to, given a distribution $\mathcal{D}$, distinguish between $\mathrm{OPT}_\gamma^{\mathcal{D}} \leq \kappa$ or $\mathrm{OPT}_{0-1}^{\mathcal{D}} >$
805    $\alpha \cdot \kappa + \Omega(\frac{1}{\mathrm{poly}(d)})$ where $\gamma = \frac{1}{d^{\mathrm{polyloglog}(d)}}$ and $\alpha = d^{1/\mathrm{polyloglog}(d)} = (1/\gamma)^{1/\mathrm{polyloglog}(1/\gamma)}$. In other
806    words, if we have a polynomial time $\alpha$-agnostic learner for $\gamma$-margin halfspaces for this regime of
807    parameter, then NP = RP.

808    Similarly, by plugging in our reduction the Sliding Scale Conjecture, we get that it is NP-hard to,
809    given a distribution $\mathcal{D}$, distinguish between $\mathrm{OPT}_\gamma^{\mathcal{D}} \leq \kappa$ or $\mathrm{OPT}_{0-1}^{\mathcal{D}} > \alpha \cdot \kappa + \Omega(\frac{1}{\mathrm{poly}(d)})$ where
810    $\gamma = 1/d^{O(1)}$ and $\alpha = d^{\Omega(1)} = (1/\gamma)^{\Omega(1)}$. In other words, if we have a polynomial time $\alpha$-agnostic
811    learner for $\gamma$-margin halfspaces for this regime of parameter, then NP = RP.    □

## Footnotes

[4]We use $\phi|_S$ to denote the restriction of $\phi$ on the domain $S$.