[Reviews · NeurIPS 2019]

Reviewer 1



The paper is of great quality, both in terms of the writing style and the technical content. One question I have, which is not very well address in the paper, is if the sample complexity upper bound of 1/\gamma^2 \epsilon^2 is optimal especially under the restricted approximation factor considered in Theorem 3.2 and Theorem 3.3, and would appreciate if the authors can shed some light on the same. There are some technical typos in the paper, which I am sure the authors will fix in the final version. Some of them are listed below: 1. Line 110, should be w^{i} in the RHS instead of w^{i+1}. 2. Like 256, incomplete parenthesis. 3. Like 268, should be 1/gamma^2 instead of 1/\gamma. 4. Line 254, by the standard arguments, it seems that the \epsilon term should contain a multiplicative factor of 1/\gamma. __________________ Post Author Feedback: Thanks for the feedback. I still strongly support to accept the paper.

Reviewer 2



This paper studies proper agnostic learning of the class of large margin halfspaces. Prior to this work the best known result of [BS00] proposed a proper learning algorithm that achieves error OPT_gamma + eps in time poly(d/eps)2^O(log(1/eps) gamma^2). The current paper shows that if one is happy with a PTAS, i.e, (1+delta)OPT_gamma + eps, then there is a proper robust learning algorithm that uses O(1/eps^2 \gamma^2) samples and runs in time poly(d/eps)2^O(1/\gamma^2) (for constant delta). The paper also shows a matching lower bound that any constant factor approximation that is a proper learner must incur exp(1/gamma) run time. Techniques: --------------- The key algorithmic insight is that if the current guess w has a large error, namely more than (1+delta)OPT then there must be many examples where w has margin less than gamma/2 whereas w*, the optimal halfspaces has margin more than gamma. This implies that w-w* has a large projection onto the matrix M = E[xx^T] where the expectation is over examples where w has low margin. Hence, in time exp(1/gamma^2) one can try all large projections onto the subspace, add them to the current vector to produce a new set of candidate vectors and proceed. One then needs to argue that the depth of this search tree is small. Overall the paper is well written and makes a solid contribution towards the understanding of proper agnostic learning of large margin halfspaces.

Reviewer 3



This paper gives a new algorithm for properly agnostically learning halfspaces with a misclassification error that is epsilon-far from (almost) the optimal gamma-margin error (up to some small approximation factor), with a run time that is poly(d/epsilon)2^(1/gamma^2). This result can be compared to previous work, which gave an algorithm that is poly(d) (1/epsilon)^(1/gamma^2), but gave an algorithm with no approximation factor. The proposed algorithm uses an interesting iterative approach, wherein in each stage, a projection that is aligned with the difference between the current halfspace and the optimal half space is sought, using the part of the distribution which incurs a low margin under the current half space. The paper also provides lower bounds on the computational complexity of any proper learning algorithm, under some standard complexity assumptions. These lower bounds show that the dependence on \gamma in the proposed algorithm is essentially tight. They also show that if one removes the approximation factor completely, then it is not possible to obtain a similar result, i.e. one with a polynomial dependence on $d$ and $1/epsilon$. Overall, the paper is well-written, and the results appear to be important and interesting. Detailed comments: - In section 1.4, Chow parameters are mentioned, but they are not subsequently used in the presented analysis, they are only used in the supplementary material. It is not clear how the algorithm and analysis in the supplementary relate to the ones in the paper. - Page 3, line 113: f and x_i are not defined. - Page 4, line 161: "Ruling out alpha-agnostic learners are slightly more complicated", "are" ==> "is" - Page 5, display equation in Claim 2.2: P[D'] should be P[y<= gamma/2] - Page 7, line 256: an extra parenthesis in the math formula. - Page 8, line 306: "learners learners"

[Author Response · NeurIPS 2019]

We thank the reviewers for their careful consideration of our paper and their positive feedback. In the following
paragraphs, we address some comments and questions asked by the reviewers.

## 3 Reviewer 1

Thank you very much for your review and suggestions.

**Question:** *Is the sample complexity upper bound of $1/\gamma^2\epsilon^2$ optimal especially under the restricted approximation*
*factor considered in Theorem 3.2 and Theorem 3.3?*

**Answer:** Theorem 3.2 considers the agnostic version of the problem (what we call 1-agnostic in our paper). This is
the most stringent notion of approximation and it is known that the sample complexity of this problem is $\Omega(1/(\epsilon^2\gamma^2))$
(information-theoretic lower bound). See, e.g., [BS00] or [SSS09] for an explicit reference.

Regarding weaker notions of approximation (like the ones addressed in our Theorems 3.1, 3.3 and our upper bounds in
the supplementary material) we note the following: For a constant factor approximation ratio $\alpha$, one can show that
indeed the $\Omega(1/(\epsilon^2\gamma^2))$ bound applies as well. Since we could not find an explicit reference of this fact, we remark
the following: Even for the basic case that the data is *linearly separable with margin* $\gamma$ (i.e., $\text{OPT}_\gamma^{\mathcal{D}} = 0$), the sample
complexity of learning is $\Omega(1/(\epsilon\gamma^2))$. This is a known fact and can be found, e.g., in Cristianini Shawe-Taylor's book.
Therefore, even for weaker notions of approximation (i.e. $\alpha = \infty$) our sample complexity is optimal as a function
of $\gamma$ and within a quadratic of optimal as a function of $\epsilon$. Note that previous algorithms (with similar approximation
guarantees) had sample complexity exponential in $1/\gamma$.

**Question:** *Line 254, by the standard arguments, it seems that the $\epsilon$ term should contain a multiplicative factor of $1/\gamma$.*

**Answer:** We are not sure exactly what the reviewer means here. Fact 2.4 (lines 251-254) is a standard generalization
bound that we quoted from the literature, saying that after $\Omega(1/(\epsilon^2\gamma^2))$ samples the empirical distribution is accurate
with high constant probability. In particular, we believe that line 254 is correct as is.

We will fix the other typos in the revised version of the paper.

## 23 Reviewer 2

Thank you very much for your review and suggestions.

## 25 Reviewer 3

Thank you very much for your review and suggestions.

**Question:** *In section 1.4, Chow parameters are mentioned, but they are not subsequently used in the presented analysis,*
*they are only used in the supplementary material. It is not clear how the algorithm and analysis in the supplementary*
*relate to the ones in the paper.*

**Answer:** The reviewer is correct that the Chow parameters do not appear again in the main body of the paper. We use
the notion of Chow parameters in our second algorithm for large values of $\alpha$ (Theorem 1.2), and its proof is entirely in
the supplementary material. This algorithm is based on a somewhat different approach than our main algorithmic result
(Theorem 2.1). We will make sure to highlight this more clearly in the revised version.

A brief high-level description of our algorithm for large values of $\alpha$ can be found at the end of page 3. To reiterate here,
it is well-known that the Chow parameters of a halfspace uniquely determine the function. With a margin condition, one
can show a robust version of this statement. Our algorithm attempts to approximate the true Chow parameters. In the
realizable setting, this would amount to simply using the empirical Chow parameters. However, in the agnostic setting
we consider, the corruptions can only produce a large change in the empirical Chow parameters if the error points in
question are very large in the direction of the change. Our algorithm works by guessing a small number of points and
guessing a correction for the empirical Chow parameters in the subspace spanned by them. Once we have robustly
learned the Chow parameters, we can use them to efficiently find the weights of an approximate linear separator by
leveraging known techniques.

We will fix the other typos pointed out in the revised version of our paper.

[Meta-Review · NeurIPS 2019]

This work makes significant progress on a relatively natural theoretical problem of proper agnostic learning halfspaces with a margin. The paper is technically strong and is a clear accept. A downside is that the problem and proposed algorithms appear to be largely irrelevant to practice.